# Cross-stage neural pattern similarity in the hippocampus predicts false memory derived from post-event inaccurate information

Xuhao Shao [1,2,3,4], Ao Li[1], Chuansheng Chen[5], Elizabeth F. Loftus[5] & Bi Zhu [1,2,3,4] ✉

The misinformation effect occurs when people's memory of an event is altered by subsequent inaccurate information. No study has systematically tested theories about the dynamics of human hippocampal representations during the three stages of misinformation-induced false memory. This study replicates behavioral results of the misinformation effect, and investigates the cross-stage pattern similarity in the hippocampus and cortex using functional magnetic resonance imaging. Results show item-specific hippocampal pattern similarity between original-event and post-event stages. During the memory-test stage, hippocampal representations of original information are weakened for true memory, whereas hippocampal representations of misinformation compete with original information to create false memory. When false memory occurs, this conflict is resolved by the lateral prefrontal cortex. Individuals' memory traces of post-event information in the hippocampus predict false memory, whereas original information in the lateral parietal cortex predicts true memory. These findings support the multiple-trace model, and emphasize the reconstructive nature of human memory.

False memory is a phenomenon in which an individual's memory of an event differs from the way it actually occurred[1-4]. It can result from the misinformation effect (i.e., distorted memory of an event after subsequent exposure to misleading information). Because of its significant implications for the legal, health and educational systems, the misinformation effect has been extensively studied for the past 50 years[5-8].

Human beings are not alone in experiencing the misinformation effect. It has been observed in a variety of nonhuman species, including fruit flies, rodents, birds, and chimpanzees[8-12]. For example, when contextual memory is reactivated using optogenetics and associated with electric shocks, false memory is created in the hippocampus of mice[13]. Researchers have speculated that the misinformation effect is a byproduct of the evolutionary adaptation of associative memory[14,15].

The hippocampus appears to be critically involved in the misinformation effect, but the dynamics of hippocampal representations across stages are still unclear[1,16]. In the classic three-stage misinformation paradigm, a person witnesses the original event, then receives post-event misinformation, and finally performs a memory test about the original event. Previous functional neuroimaging studies have shown that the human hippocampus is activated in each of these three memory stages to produce false memory, although these results are occasionally not replicated[17-22]. These inconsistent results may be due to the fact that previous studies used different experimental designs and investigated hippocampal activity in only one or two memory stages. Using representational similarity analysis (RSA), recent studies have shown that the human hippocampus supports false memory for recombined images[23], and constructs narratives

[1]State Key Laboratory of Cognitive Neuroscience and Learning, Beijing Normal University, 100875 Beijing, China. [2]Institute of Developmental Psychology, Beijing Normal University, 100875 Beijing, China. [3]Beijing Key Laboratory of Brain Imaging and Connectomics, Beijing Normal University, 100875 Beijing, China. [4]IDG/McGovern Institute for Brain Research, Beijing Normal University, 100875 Beijing, China. [5]Department of Psychological Science, University of California, Irvine, CA 92697, USA. ✉e-mail: zhubi@bnu.edu.cn

across distant events[24]. However, it is largely unknown how human hippocampal representations change across three memory stages to create false memory of life events.

Three theoretical perspectives have been proposed to explain the misinformation effect: non-retention, trace-alteration, and multiple-trace models (see Fig. 1 for a schematic representation of these models). These theoretical perspectives would predict different patterns of changes in hippocampal representations across the three stages of misinformation false memory. First, the non-retention model argues that representations of the original event were never formed or were lost before receiving misinformation, and thus the original event was filled with misinformation[25]. According to this model, the hippocampal pattern from the original-event stage would not be related to those during the post-event stage (i.e., no similarity), but its pattern of the post-event stage and the memory-test stage would be related (Fig. 1a).

Second, the memory trace-alteration model proposes that hippocampal representations of the original event are retained during the post-event stage but are then overwritten by misinformation, and consequently do not carry over to the memory-test stage[2,4,26]. Therefore, the hippocampal pattern for the original event should carry over from the original-event stage to the post-event stage, but would then be replaced by misinformation, whereas its pattern for the misinformation would carry over from the post-event stage to the memory-test stage (Fig. 1b).

Finally, the multiple-trace memory model holds that the hippocampal representations of the original event remain intact throughout the three stages, and they compete with the representations of misinformation during the memory-test stage[27]. The outcome of the competition is determined by source attribution, with source misattribution leading to false memory[3,28]. According to this model, the hippocampal representation of the original event should exist across the three stages and the representation of misinformation would carry from the post-event stage to the memory-test stage. In other words, unlike the other two perspectives, the multi-trace model would predict the existence of hippocampal representations of the original event during the memory-test stage even if misinformation was effective in creating false memory (Fig. 1c). A key mechanism involved here is cognitive control, which should be subserved by certain parts of the prefrontal cortex[29,30]. To resolve the competition between two memory traces in the hippocampus during the memory test, the ventrolateral, medial, and dorsolateral prefrontal cortex may be involved in selecting goal-relevant memory of original information[23,31], suppressing inappropriate memory of misinformation[32,33], and monitoring the discrepancy between misinformation and original information[34,35]. However, it was unclear which parts of the prefrontal cortex might work in concert with the hippocampus to resolve this conflict during the memory test.

The current study aimed to test the three competing theories of the misinformation effect in terms of hippocampal representations. Two experiments were conducted. First, we conducted a behavioral study (Exp. 1) to ensure that the misinformation paradigm we modified for the neuroimaging study would work as expected. Based on prior works (see Supplementary Table 1 for a brief review), we developed a new set of experimental materials based on realistic events in today's society. We used eight events (e.g., stealing a cell phone, ticket scalping, and school bullying), in which two versions of critical items for each event were counterbalanced across participants (Fig. 2a and Supplementary Fig. 1). Participants first saw all eight events (i.e., the original-event stage), with 50 images for each event, which took about half an hour in total (Methods and Fig. 2b). Three hours later, they read item-specific post-event narratives (i.e., the post-event stage). During this stage, participants were randomly assigned into one of three groups: misinformation, neutral, and consistent information groups (Supplementary Table 2). Participants in the misinformation group read misinformation for critical items and consistent information for control items. Participants in the neutral information group read neutral information for both critical and control items. Participants in the consistent information group read consistent information for both critical and control items. One day later (during the memory-test stage), all participants performed the item-specific three-alternative forced-choice recognition test for the original events (true, false, and foil options for critical items; and one correct and two incorrect options for control items) (Supplementary Table 3). Unlike previous studies, we used a randomized order of presentation across the eight events in each of all three stages, and across the 24 critical and control questions of each event in the memory test (Supplementary Figs. 2–3). This randomization helped to avoid potential confounding effects of temporal drift and sequence-related structure on the neural pattern similarity estimation[36].

After Exp. 1 confirmed that the modified paradigm generated expected behavioral results (i.e., the misinformation group led to more false memory than did the neutral and consistent groups), we recruited a new sample to examine the hippocampal representations (Exp. 2). In this fMRI experiment, we included only the misinformation group. Participants completed all three stages of the misinformation paradigm in the fMRI scanner. Exp. 2 used a within-participant design and examined the misinformation effect by comparing behavioral and neural indices between memory response types (i.e., true memory, false memory, foil, correct control, and incorrect control). Using representational similarity analyses of fMRI data, we compared the hippocampal pattern similarity across three stages and tested predictions derived from the three theories mentioned above. Our results showed that cross-stage neural pattern similarity in the hippocampus predicted the misinformation effect.

## Results
### Behavioral results
The misinformation effect was demonstrated by group differences in memory performance in the behavioral study (Exp. 1; Fig. 2c). There was a significant interaction between group (misinformation, neutral, and consistent) and memory type (true, false, foil, correct, and incorrect) on the endorsement rate in the memory test ($F(8, 476) = 91.62$, $p = 8e^{-43}$, $\eta^2_p = 0.61$, 90% confidence intervals (CI) = [0.56, 0.64]). As expected, true memory (i.e., endorsement rate of original information) in the misinformation group was lower than that in the neutral group ($t(80) = -6.53$, $p = 6e^{-9}$, Cohen's $d = -1.44$, 95% CI = [−1.93, −0.95]),

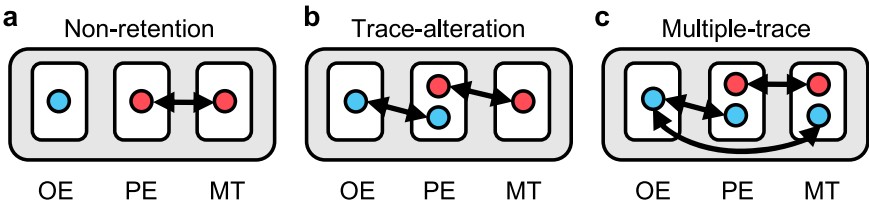

**Fig. 1 | Three theoretical models of the misinformation effect. a** The non-retention model. **b** The trace-alteration model. **c** The multiple-trace model. They posit different patterns of changes in hippocampal representations across the three stages of misinformation false memory (original-event [OE], post-event [PE], and memory-test [MT] for the original event). Blue and red spheres indicate representations of original information and post-event misinformation, respectively. Double-headed arrows indicate cross-stage neural pattern similarity for the corresponding items.

**a   Experimental Materials**

**Eight events**: Steal, scalper, gamble, rob, bully, scam, loan shark, and family conflicts

**Event 1:**  (A man steals a girl's phone on the street)

Two versions of critical items for each event were counterbalanced across participants

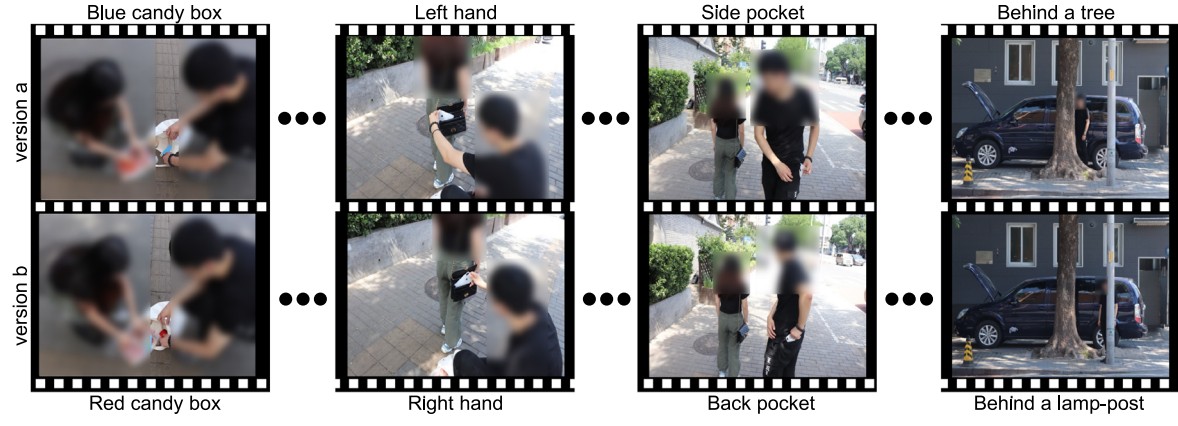

**b   Three-stage task (examples of critical and control items)**

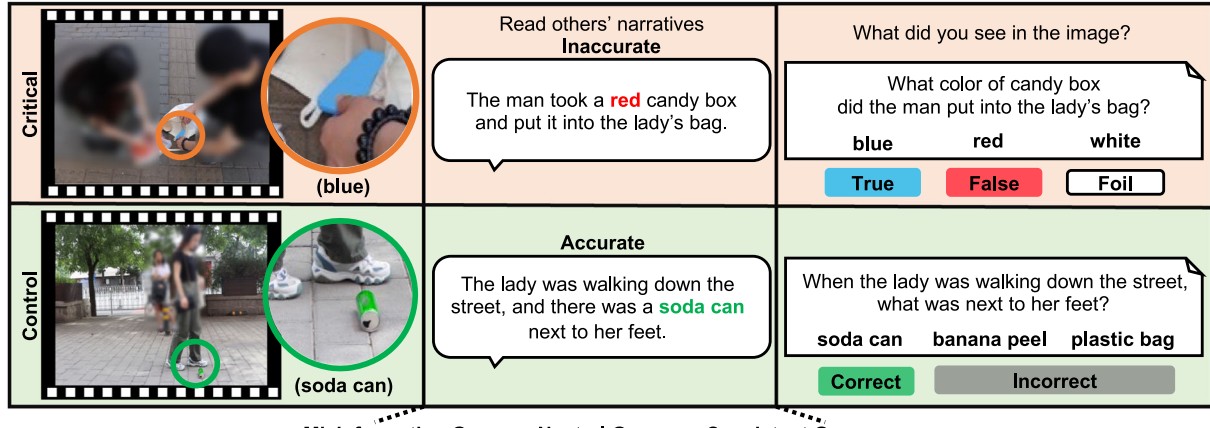

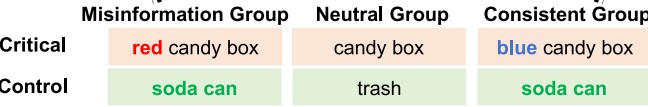

**c   Memory performance**

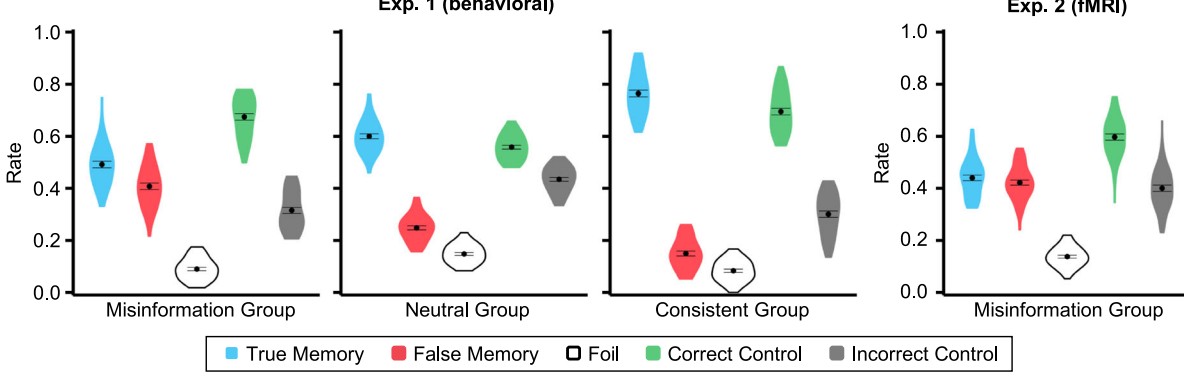

**Fig. 2 | Experimental design and behavioral results of the misinformation effect. a** Materials of original events. Images of people in the figure are blurred according to journal's regulations, while images in the experiment were displayed in high resolution. **b** Procedure and manipulation. Critical items are shown in light orange and control items are shown in light green. True memory, false memory, foil, correct control, and incorrect control are shown in blue, red, white, dark green, and gray, respectively. **c** Memory performance. Data are visualized as violin plots with bounds indicating the min and max values and the black dots showing the means. Error bars indicate standard errors (SEs) of the means for experiment (Exp.) 1 and within-participant SEs for Exp. 2. $N = 122$ participants in Exp. 1. $N = 57$ participants in Exp. 2. The two experiments used independent samples. Source data are provided as a Source Data file. Permission has been obtained from these individuals to depict them in this figure.

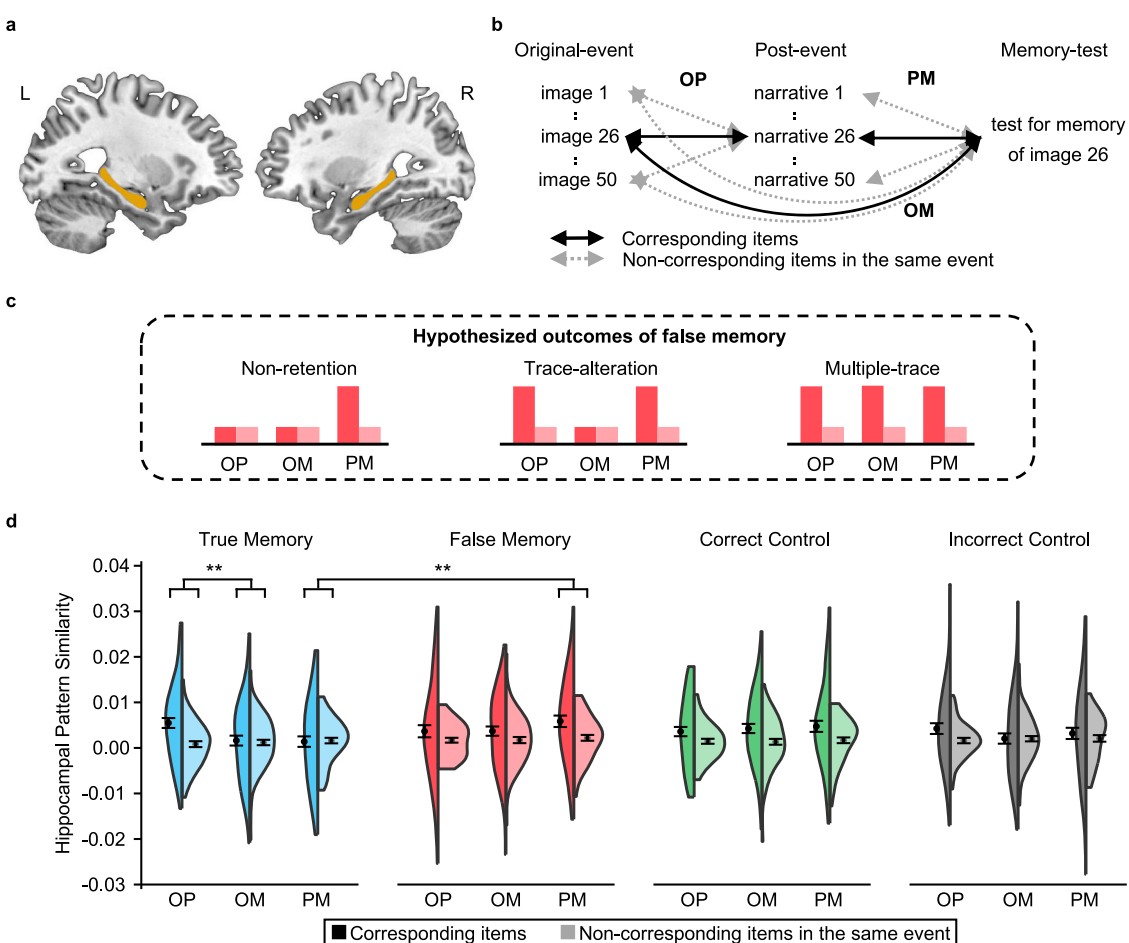

**Fig. 3 | Hippocampal pattern similarity across three memory stages. a** The hippocampus (yellow) on the left (L) and right (R) hemispheres of the brain. **b** Three types of cross-stage similarity (i.e., OP [between original-event and post-event stages], OM [between original-event and memory-test stages], and PM [between post-event and memory-test stages]) for the corresponding items and for the non-corresponding items in the same event. **c** Hypothesized outcomes of false memory based on the three theories. **d** Hippocampal pattern similarity for the corresponding items and for the non-corresponding items for three different stage pairs by memory type (true memory, false memory, correct control, and incorrect control) in Exp. 2 ($N = 57$). Paired sample $t$-test showed that the effect of item specificity (i.e., corresponding items vs. non-corresponding items) for true memory was greater in OP than in OM ($t(56) = 3.06$, $p = 0.003$, $d = 0.41$, 95% CI = [0.13, 0.67], two-sided, FDR corrected $p = 0.005$). The effect of item specificity in PM was greater for false memory than for true memory ($t(56) = 2.85$, $p = 0.006$, $d = 0.38$, 95% CI = [0.11, 0.64], two-sided, FDR corrected $p = 0.036$). Data are visualized as split-violin plots with bounds indicating the min and max values and the black dots showing the means. Error bars indicate within-participant SEs. **$p < 0.01$. Source data are provided as a Source Data file.

which in turn was lower than that in the consistent group ($t(80) = -10.19$, $p = 4e^{-16}$, $d = -2.25$, 95% CI = [−2.80, −1.69]). False memory (i.e., endorsement rate of misinformation) in the misinformation group was higher than that in the neutral group ($t(80) = 11.09$, $p = 2e^{-16}$, $d = 2.45$, 95% CI = [1.87, 3.02]), which in turn was higher than that in the consistent group ($t(80) = 8.19$, $p = 3e^{-12}$, $d = 1.81$, 95% CI = [1.29, 2.32]). The misinformation effect was further confirmed in both experiments. Within the misinformation group in Exp. 1, false memory (41%) was higher than the endorsement rate of foils (9%) ($t(39) = 21.29$, $p = 4e^{-32}$, $d = 3.37$, 95% CI = [2.56, 4.17]). The behavioral results from the fMRI study (Exp. 2) replicated those of the misinformation group: false memory (42%) was higher than the endorsement rate of foils (14%) ($t(56) = 27.01$, $p = 8e^{-34}$, $d = 3.58$, 95% CI = [2.87, 4.29]). Detailed results on endorsement rates and reaction time from both experiments are shown in the Supplementary Information (Supplementary Tables 4–5 and Supplementary Fig. 4).

## Hippocampal representations of true, false, correct, and incorrect memories

To explore hippocampal representations of different types of memories, we looked for evidence of item-specific representation in the bilateral hippocampus (Fig. 3a). Item-specific representation was assessed for each type of memory (i.e., true memory, false memory, correct control, and incorrect control) and for each pair of the three stages: between the original-event and the post-event (OP), between the original-event and the memory-test (OM), and between the post-event and the memory-test (PM) (Fig. 3b). For a given stage pair, item-specific representation was indicated by greater neural pattern similarity for the corresponding items than the averaged neural pattern similarity for the non-corresponding items in the same event (Supplementary Fig. 5). Taking a critical item of the OP as an example, the corresponding items would be seeing "the man took a blue candy box" (image 26) during the original-event stage and reading "the man took a red candy box" (narrative 26) during the post-event stage; while the non-corresponding items in the same event would be seeing the other 49 images (except for image 26) during the original-event stage and reading "the man took a red candy box" (narrative 26) during the post-event stage.

Results showed that for some stage pairs of the four memory types, hippocampal pattern similarity was higher for corresponding items than for non-corresponding items, suggesting evidence for item-specific representation and that it differed by memory type for different stage pairs (Fig. 3d). This observation was confirmed by a significant three-way interaction of the 3 (stage pairs) × 4 (memory

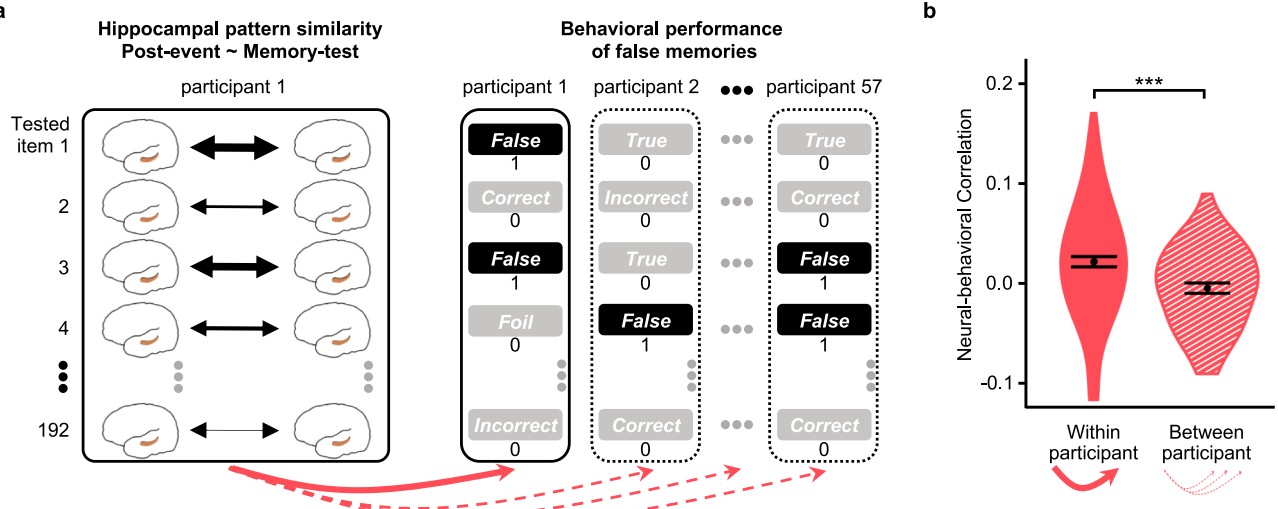

**Fig. 4 | Hippocampus carries participant-specific post-event information that predicts false memory. a** Schematic representation of the individuation analysis for the correlation between the hippocampal item-specific representations of post-event information and the behavioral pattern of false memories in the memory test. We compared the within-participant neural-behavioral correlations (red solid arrow) to the between-participant neural-behavioral correlations (red dashed arrows). **b** The neural-behavioral correlations for false memory are shown for the within- and between-participant analyses in Exp. 2 ($N = 57$). Paired sample $t$-test showed that within-participant correlation was higher than between-participant correlation ($t(56) = 3.64$, $p = 0.0006$, $d = 0.48$, 95% CI = [0.21, 0.75], two-sided). Data are visualized as violin plots with bounds indicating the min and max values and the black dots showing the means. Error bars indicate within-participant SEs. ***$p < 0.001$. Source data are provided as a Source Data file.

types) × 2 (item specificity) repeated measures ANOVA ($F(6, 336) = 2.27$, $p = 0.036$, $\eta^2_p = 0.04$, 90% CI = [0.001, 0.06]), and a significant main effect of item specificity ($F(1, 56) = 27.82$, $p = 2e^{-6}$, $\eta^2_p = 0.33$, 90% CI = [0.17, 0.47]), while the other main effects and the two-way interactions were not significant in this model ($ps > 0.13$).

To probe the three-way interaction, we conducted two sets of two-way ANOVAs, by memory type and stage pair. In terms of true memory, the two-way interaction between stage pair and item specificity was significant ($F(2, 112) = 6.77$, $p = 0.002$, $\eta^2_p = 0.11$, 90% CI = [0.03, 0.20]). Simple effect analysis showed that the hippocampal pattern similarity of true memory was higher for corresponding items than for non-corresponding items in OP ($t(56) = 4.33$, $p = 6e^{-5}$, $d = 0.57$, 95% CI = [0.29, 0.85]) but not in OM or PM ($ps > 0.65$), and the effect of item specificity was greater in OP than in OM and PM ($t(56) = 3.06$ and 3.19, $p = 0.003$ and 0.002, $d = 0.41$ and 0.42, 95% CI = [0.13, 0.67] and [0.15, 0.69]). For false memory, the effect of item specificity was significant ($F(1, 56) = 16.40$, $p = 0.0002$, $\eta^2_p = 0.23$, 90% CI = [0.08, 0.37]), but the effect of stage pair and the interaction between stage pair and item specificity were not significant ($ps > 0.44$). These results were consistent with the multiple-trace theory (Fig. 3c). For correct control, the effect of item specificity was significant ($F(1, 56) = 17.37$, $p = 0.0001$, $\eta^2_p = 0.24$, 90% CI = [0.09, 0.38]), but the effect of stage pair and the interaction between stage pair and item specificity were not significant ($ps > 0.74$). Finally, for incorrect control, none of the effects (stage pair, item specificity, or their interaction) was significant ($ps > 0.09$).

In terms of the two-way ANOVAs by stage pair, there was a significant two-way interaction between memory type and item specificity for PM ($F(3, 168) = 2.84$, $p = 0.040$, $\eta^2_p = 0.05$, 90% CI = [0.001, 0.10]). Simple effect analysis showed that hippocampal pattern similarity in PM was higher for corresponding items than for non-corresponding items for false memory and correct control ($t(56) = 3.62$ and 2.83, $p = 0.0006$ and 0.006, $d = 0.48$ and 0.38, 95% CI = [0.20, 0.75] and [0.11, 0.64]), but not for true memory and incorrect control ($ps > 0.41$). The effect of item specificity in PM was greater for false memory and correct control than for true memory ($t(56) = 2.85$ and 2.20, $p = 0.006$ and 0.032, $d = 0.38$ and 0.29, 95% CI = [0.11, 0.64] and [0.02, 0.56]). In contrast, for OP or OM, the effect of item specificity was significant ($F(1, 56) = 19.98$ and 8.94,

$p = 0.00004$ and 0.004, $\eta^2_p = 0.26$ and 0.14, 90% CI = [0.11, 0.41] and [0.03, 0.28]), but the effect of memory type and the interaction between memory type and item specificity were not significant ($ps > 0.15$). In addition, we directly compared false memory with correct control by conducting a 2 (memory types: false memory vs. correct control) × 3 (stage pairs) × 2 (item specificity) repeated measures ANOVA. Since none of the interaction terms was significant, we combined false memory and correct control and compared them with true memory using a 2 (memory types: [false memory & correct control] vs. true memory) × 3 (stage pairs) × 2 (item specificity) repeated measures ANOVA. These analyses confirmed that the effect of item specificity in PM in the hippocampus was greater for false memory and correct control than for true memory (Details are shown in Supplementary Table 6). Furthermore, the results described above were not affected by the univariate activation level in the hippocampus, and were unchanged when using partial correlations for PM and OM and after correction for correlation comparisons (Details are shown in Supplementary Tables 6–7). Additional results regarding the anterior and posterior hippocampus indicated a functional dissociation along the long axis of the hippocampus for true memory and correct control but not for false memory and incorrect control (Details are shown in Supplementary Table 8).

## Hippocampal representation of post-event information predicts false memory

The analyses reported above suggested that there were two memory traces (i.e., significant OM and PM values) in the hippocampus for false memory and correct control during the memory test. Furthermore, item-specific representation of post-event information (i.e., PM) in the hippocampus was greater for false memory and correct control than for true memory. However, it was unclear whether there would be a participant-specific mapping between the hippocampal item-specific representation of post-event information and the behavioral pattern of false memory. Using an individuation analysis as used in previous studies[37,38], we compared the within-participant neural-behavioral correlations (unique representations) to the between-participant neural-behavioral correlations (shared representations) (Fig. 4a). For false memory, this analysis revealed a significantly higher within- than

**a**

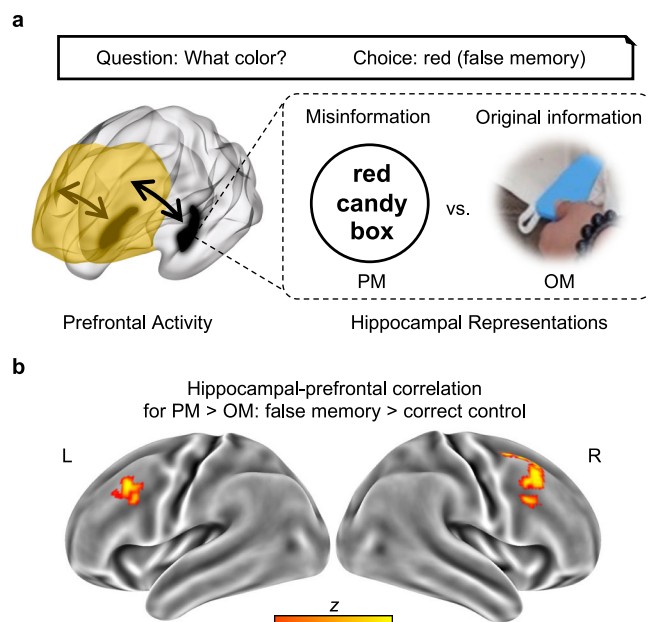

**b**

Hippocampal-prefrontal correlation
for PM > OM: false memory > correct control

**Fig. 5 | Prefrontal activity correlates with hippocampal representations when false memory occurs. a** The hypothesis was that prefrontal activity would be more positively correlated with hippocampal representation of post-event misinformation (PM) than that of original information (OM) when false memory occurred during the memory test. The prefrontal cortex is shown in yellow, and the hippocampus is shown in black. Permission has been obtained from the individual to appear in this figure. **b** The lateral prefrontal activity showed a more positive correlation with hippocampal item-specific representation of post-event information than with that of original information, and this effect was more pronounced for false memory than for correct control during the memory test in Exp. 2 ($N = 57$).

between-participant correlation ($t(56) = 3.64$, $p = 0.0006$, $d = 0.48$, 95% CI = [0.21, 0.75]; Fig. 4b and Supplementary Table 9), suggesting that each individual had a partially unique set of representations of post-event information in the hippocampus to produce false memory. However, there was no participant-specific mapping between the hippocampal representation of the original information during the memory test and the behavioral pattern of false memory ($t(56) = 1.21$, $p = 0.23$, $d = 0.16$, 95% CI = [−0.10, 0.42]). There was also no participant-specific mapping between the hippocampal item-specific representation of the original or post-event information during the memory test (i.e., OM or PM) and the behavioral pattern of correct control ($t(56) = 1.41$ and 0.80, $p = 0.16$ and 0.43, $d = 0.19$ and 0.11, 95% CI = [−0.08, 0.45] and [−0.15, 0.37] for OM and for PM, respectively). In the hippocampus, there was no participant-specific mapping between OM and true memory ($t(56) = −1.55$, $p = 0.13$, $d = −0.21$, 95% CI = [−0.47, 0.06]), but the magnitude of negative correlation between PM and true memory was significantly smaller in within-participant than between-participant analyses ($t(56) = −2.32$, $p = 0.024$, $d = −0.31$, 95% CI = [−0.57, −0.04], Supplementary Table 9 for details).

**Prefrontal activity correlates with hippocampal representations when false memory occurs**

In support of the multiple-trace model, the above analysis suggested that there were two memory traces (i.e., OM and PM) in the hippocampus when false memory occurred during the memory test. Given the two memory traces, the prefrontal cortex is expected to be involved in selecting, suppressing, or monitoring memory traces in the hippocampus during the memory test (Fig. 5a). When false memory occurs, the neural activity in certain regions of the prefrontal cortex may be positively correlated with OM (i.e., selecting original information), or negatively correlated with PM (i.e., suppressing

misinformation), or more positively correlated with PM than OM in the hippocampus (i.e., monitoring the discrepancy). Moreover, this effect should be more pronounced for false memory than for correct control, since hippocampal item-specific representations in OM and PM for correct control involve consistent information. We conducted these analyses for false memory and correct control rather than for true memory and incorrect control, because there were significant hippocampal item-specific representations in OM and PM for false memory and correct control only.

Using an exploratory whole-brain analysis and controlling for the level of hippocampal activity during the memory test, we identified two clusters located in the left lateral prefrontal cortex (MNI, −42, 28, 44, $Z = 3.71$, cluster size = 170) and the right lateral prefrontal cortex (MNI, 56, 26, 36, $Z = 3.81$, cluster size = 263; Fig. 5b). They met the requirements that their activity showed a more positive correlation with the hippocampal item-specific representation in PM than that in OM for false memory, and that the effect described above was greater for false memory than for correct control. For the left lateral prefrontal cortex, hippocampal-prefrontal correlations for false memory were positive in PM ($t(56) = 2.41$, $p = 0.019$, $d = 0.32$, 95% CI = [0.05, 0.58]) and negative in OM ($t(56) = −2.41$, $p = 0.019$, $d = −0.32$, 95% CI = [−0.58, −0.05]), whereas hippocampal-prefrontal correlations for correct control were non-significant in either PM or OM ($ps > 0.15$). For the right lateral prefrontal cortex, hippocampal-prefrontal correlations for false memory were positive at trend level in PM ($t(58) = 1.82$, $p = 0.074$, $d = 0.24$, 95% CI = [−0.02, 0.50]) and negative in OM ($t(56) = −2.75$, $p = 0.008$, $d = −0.36$, 95% CI = [−0.63, −0.10]), whereas hippocampal-prefrontal correlations for correct control were negative in PM ($t(56) = −2.55$, $p = 0.013$, $d = −0.34$, 95% CI = [−0.60, −0.07]) but not in OM ($p = 0.42$). Besides the prefrontal cortex, there was a cluster in the left superior parietal lobe (MNI, −40, −64, 26, $Z = 4.23$, cluster size = 408). No brain region showed the reversed pattern (i.e., a more positive hippocampal-prefrontal correlation in OM than that in PM for false memory compared with correct control).

**Cortical representations and their connectivity with hippocampus**

The current study focused on hippocampal representations underlying the misinformation effect. However, it is also possible that cortical representations were involved in this process. Thus, we explored whether there were other brain regions showing differences between true and false memories in item-specific representations of OM and PM. Based on the whole-brain searchlight analysis, multiple brain regions (e.g., the left angular gyrus) showed greater item-specific representations in OM for true memory than false memory, but only the posterior cingulate gyrus showed greater item-specific representations in PM for false memory than true memory (Fig. 6a). Next, we explored which brain regions carried participant-specific representations of original information to produce true memory or carried misinformation to produce false memory using the individuation analysis. Based on the whole-brain analysis, the lateral parietal cortex (e.g., the left angular gyrus) showed a participant-specific neural-behavioral correlation for true memory, but the medial parietal cortex (e.g., the posterior cingulate gyrus) showed a participant-specific neural-behavioral correlation for false memory (Fig. 6b and Supplementary Table 10 for details). Additional results on cortical representations and activations are shown in the Supplementary Information (Supplementary Figs. 6–11 and Supplementary Tables 11–16).

To explore the potential informational connectivity[39,40], we assessed the covariation in trial-by-trial information (i.e., OM or PM) between the hippocampus and the cortex during the memory test. Specifically, we tested whether their trial-by-trial correlation coefficients in OM were higher for true memory than false memory, while their trial-by-trial correlation coefficients in PM were higher for false memory than true memory. Based on the whole-brain analysis with the

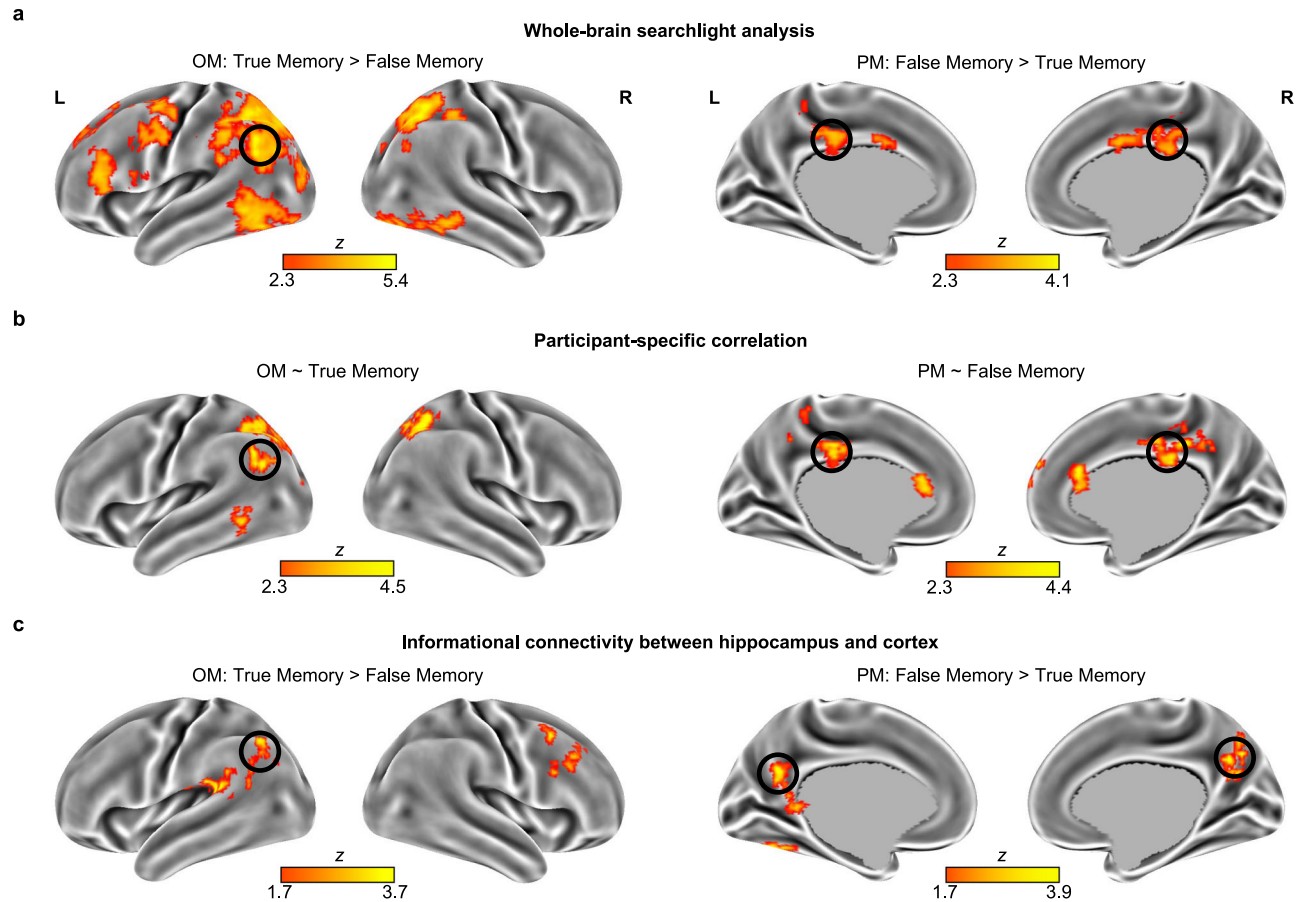

**Fig. 6 | Cortical representations of true and false memories and their connectivity with the hippocampus. a** Multiple brain regions (e.g., the left angular gyrus) showed greater item-specific representations of original information during the memory test (OM) for true memory than false memory, but only the posterior cingulate gyrus showed greater item-specific representations of post-event information during the memory test (PM) for false memory than true memory. **b** Participant-specific neural-behavioral correlation. The lateral parietal cortex (e.g., the left angular gyrus) carried participant-specific original information that predicted true memory, whereas the medial parietal cortex (e.g., the posterior cingulate gyrus) carried participant-specific misinformation that predicted false memory. **c** Informational connectivity between the hippocampus and the cortex. The correlation coefficients between the hippocampus and several cortical regions (e.g., the left angular gyrus) in OM were higher for true memory than false memory, whereas the correlation coefficients between the hippocampus and the precuneus (extending to the posterior cingulate gyrus) in PM were higher for false memory than true memory in Exp. 2 ($N = 57$).

hippocampus as a seed, the correlation coefficients between the hippocampus and several cortical regions (e.g., the left angular gyrus) in OM were higher for true memory than false memory, but the correlation coefficients between the hippocampus and the precuneus (extending to the posterior cingulate gyrus) in PM were higher for false memory than true memory (Fig. 6c). Additional brain regions revealed by this analysis are shown in Supplementary Table 17.

## Discussion

Behavioral results of both experiments of this study demonstrated an overall misinformation effect, which is consistent with hundreds of previous studies[4,8,22,41–43]. In Exp. 1, participants had the highest rate of false memory after reading misinformation, but the lowest rate of false memory after reading consistent information, and a rate somewhere in between after reading neutral information. Moreover, participants in the misinformation group of both experiments had higher false memory for misinformation (~40%) than false alarms for foils (~10%).

After demonstrating the misinformation effect using our adapted experimental design, we used fMRI and representational similarity analysis in Exp. 2 to examine how the human hippocampus created false memory as a result of the misinformation effect. We first showed that the hippocampus carried corresponding original information during the post-event stage. Previous univariate studies have explored the differences in neural activation during the original and post-event stages for subsequent true and false memories, which are thought to reflect item-encoding and source-encoding processes in the hippocampus[17,22]. For the post-event stage with extra manipulations (e.g., re-displaying and comparing the original information and misinformation), prior studies have artificially induced reactivation, showing that hippocampal activity during this induced reactivation supports subsequent false memories[18,21]. Extending previous univariate studies, our study found hippocampal pattern similarity between original and post-event stages, which suggests spontaneous reactivation of corresponding original information in the hippocampus during the post-event misinformation stage.

This finding refutes the non-retention interpretation of the misinformation effect, which claims that memories of original events were lost prior to reading misinformation[25]. On the contrary, our finding suggests that when people read misinformation, the representation of corresponding original information could be retrieved and revived in the hippocampus. Specifically, when a person reads a piece of misinformation (e.g., the narrative showing "the man took a red candy box"), his/her hippocampus reactivates the pattern of activity recorded when he/she saw the corresponding image (e.g., the image showing "the man took a blue candy box"), more than the pattern of activity recorded when he/she saw the other 49 images from the same event. During the post-event stage, the hippocampus is involved in retrieving corresponding information from the original event, rather

than simply encoding misinformation. The misinformation may either harm or help memory by reactivating original information[41]. In other words, during this post-event stage, the hippocampus re-encodes the content associated with the post-event (mis)information, allowing the original memory to be weakened, enhanced, or updated[21,44]. It demonstrates the critical role of the hippocampus in memory for reactivating, binding, and updating relational information[23,45,46].

When participants took the memory test one day later, the representation of original information was found to be weakened in the hippocampus for true memory, whereas the hippocampus seemed to reinstate both original information and post-event (mis)information for false memory and correct control. In other words, the item-specific post-event misinformation was reinstated to a greater extent in the hippocampus for false memory compared to true memory. However, there was no activation difference between true and false memories in the hippocampus during the memory test, which is consistent with previous studies using univariate analysis[19,20]. Extending these two previous studies that used sensory reactivation paradigms (i.e., visual images and auditory narratives), our study provided direct evidence for hippocampal representations of two visual sources for each item (i.e., visual images and written narratives), by measuring the hippocampal pattern similarity across stages.

Our findings support the multiple-trace memory theory, which posits that two opposing memory traces of the original and misleading information coexist in the hippocampus[3,27,28,44]. When false memory was produced during the memory test, the hippocampus carried not only the corresponding post-event misinformation, but also the corresponding original information. Since there were two memory traces for false memory during retrieval, this discrepancy would trigger the source monitoring process in the lateral prefrontal cortex[29,34,47–49]. Indeed, our results suggest that the strong memory traces of misinformation but weak memory traces of original information in the hippocampus triggered monitoring processes for false memory in the lateral prefrontal cortex (i.e., positively correlated with PM but negatively correlated with OM). However, we did not observe any positive hippocampal-prefrontal correlation for OM (i.e., selecting original information), or any negative correlation for PM (i.e., suppressing misinformation). Overall, our findings indicate that the lateral prefrontal cortex works in concert with the hippocampus to resolve this conflict during the memory test when false memory occurs.

Interestingly, each individual's unique hippocampal representation of post-event information predicted idiosyncratic patterns of misinformation-induced false memory. Although one previous study found shared neural representations during encoding among participants predicted subsequent memory[50], others found that participant-specific neural representations during encoding predicted memory performance[37]. In our study, false memory was predicted by participant-specific neural representations of post-event information in the hippocampus, rather than the shared neural representations across participants.

Outside the hippocampus, an exploratory searchlight analysis showed that cortical representations also contributed to true and false memories. In terms of true memory, the frontoparietal and inferior temporal cortex carried original information during the post-event and memory-test stages, as evidenced by greater OP and OM for true memory than other memory types. However, no brain region showed greater OP and OM for false memory than true memory. These results suggest that this cross-stage neural pattern similarity is mainly due to the reactivation of the original information rather than the blurring of the two pieces of encoded information. Among these brain regions, the lateral parietal cortex carried participant-specific neural representations of original information that predicted true memory during the memory test. Unlike the typical role of the hippocampus in predicting true memory[51–54], the post-event misinformation altered the role of the hippocampus without changing the role of the lateral

parietal cortex in true memory. Although the lateral parietal cortex rather than the hippocampus carried original information during the memory test, the original information was communicated between these two brain regions, as evidenced by the stronger informational connectivity between the hippocampus and the left angular gyrus in OM for true memory than false memory.

False memory was predicted by the participant-specific representations of misinformation carried by the hippocampus and medial parietal cortex (e.g., posterior cingulate gyrus), which communicated with each other during the memory test. It should be noted that the hippocampal effect found in the ROI analysis was replicated in the whole-brain neural-behavioral correlation analysis, but not in the whole-brain searchlight analysis. One possible reason is that the searchlight method has several limitations, including mismatches in size and shape of relevant brain regions and those of the searchlight[55]. These findings were consistent with previous studies showing that lateral and medial parietal cortices differ in their role in memory representation during retrieval[56,57]. In the misinformation paradigm, true memory is associated with the neural representation in the left angular gyrus as it retrieves original-event episodic details[58–61]; whereas false memory is associated with the neural representation in the posterior cingulate gyrus as it links post-event narratives with prior knowledge[62–64]. Extending these previous studies, our study showed that each individual's unique neural representation of original or post-event information in these brain regions predicted idiosyncratic patterns of true or false memory. In line with the role of the posterior medial network[56], our findings suggest that the lateral and medial parietal cortices maintain distinct hippocampal-cortical communications of original-event and post-event information to produce true and false memories.

To mimic real-life situations, the current study had participants read post-event misinformation three hours after seeing multiple events, and take the memory test one day later. However, the time interval between encoding and retrieval may affect memory traces in the hippocampus to produce false memory[65–67]. Future studies should explore changes in hippocampal representations at longer time intervals (e.g., weeks or months) and their effects on misinformation false memory. Furthermore, our findings of the hippocampal-prefrontal correlation imply that manipulating the prefrontal activity during the memory test may influence hippocampal representations and thus reduce false memories. However, it should be noted that the hippocampal-prefrontal correlation found in this study was based on the exploratory searchlight analysis and is subject to alternative interpretations (e.g., bidirectional prefrontal-hippocampal interactions). Future studies should replicate this finding and investigate the causal role of the prefrontal activation by using behavioral interventions (e.g., warning) or brain stimulation (e.g., transcranial direct current stimulation over the prefrontal cortex)[7,68,69]. Another limitation of the current study was that we did not use cued recall tests to isolate neural patterns related to memory reactivation. In this study, the recognition test allowed for better control of the stimuli, but also enabled the reprocessing of scenes elicited by words in the memory test. However, we do not believe that perceptual reprocessing can explain our results because (a) item-specific neural pattern similarity between post-event and memory-test stages was higher for false memory than for true memory, and (b) this neural pattern similarity was associated with the behavioral performance of false memory rather than correct control. If perceptual reprocessing had been responsible for the neural pattern similarity, we would not have seen these results because both critical and control items involved the same kind of perceptual reprocessing in the memory test. Therefore, we believe that cross-stage neural pattern similarity was primarily caused by memory representations of specific details rather than by perceptual reprocessing of scenes, Nevertheless, future studies should consider using cued recall tests to avoid any potential confounding of perceptual reprocessing and to improve the ecological validity of memory

studies[24,70]. In addition, it should be noted that the results of the cortical representation were based on the exploratory searchlight analysis, which should be replicated in future studies using ROI analysis.

In conclusion, our research provides direct evidence that dynamic changes in human hippocampal representations across three memory stages underlie the misinformation effect. These neuroimaging findings support the multiple-trace memory theory and source monitoring framework of misinformation false memory. Moreover, the hippocampus works with the lateral and medial parietal cortices to produce true and false memories, respectively. These findings enhance our understanding of the neural mechanisms underlying the reconstructive nature of human memory by providing an integrated model of the hippocampal-cortical network involved in false memory.

## Methods

### Participants
In Exp. 1 (a behavioral study), 122 participants (63 females and 59 males [based on their self-report], mean age 21 ± 2 years) were randomly assigned into three groups (i.e., misinformation, neural, and consistent) (Supplementary Table 2). In Exp. 2 (an fMRI study), there was an independent sample of 57 participants for the misinformation group (29 females and 28 males [based on their self-report], mean age 22 ± 2 years). Both male and female participants were randomly assigned into these groups. These sample sizes were determined based on a power analysis of previous studies on similar topics[22,41,43], which indicated that a sample size of 30 participants for each group would reach a power over 80%. Participants are right-handed Chinese college students with normal vision and without any psychiatric or neurological disease. They were screened to ensure that they had never previously seen any material of this study. All participants provided written informed consent. They were paid 100 RMB per hour. This study was approved by the Institutional Review Board at Beijing Normal University, China.

### Materials
Images of original events, narratives for post-event information, and the memory test of eight events were developed by the authors. Eight events are about (1) a man steals a girl's phone on the street, (2) a scalper sells appointment tickets in the hospital, (3) a family fights for money while playing mahjong, (4) a little boy is robbed after running away from home, (5) a schoolboy is bullied by classmates at school, (6) a teenage boy steals money from parents to give a camgirl, (7) a female college student borrows money from loan sharks, and (8) grandfather and grandson have a conflict. Images were developed by shooting photos for the first three events, and by taking snapshots from videos on the local television stations for the remaining five events. These events involve pictures of Chinese people of different ages and regions, who are unknown to most people. The development of these materials was inspired by previous studies that used events of young Americans about 15 years ago[20,22].

For each event, 50 color digital slide images were used to describe one theme, including 12 critical items, 12 control items, and 26 generic items. Critical and control items, but not generic items, would be tested later. For critical items, they would be described inaccurately for the misinformation group in the subsequent post-event stage. To increase the credibility of narratives, critical and control items would not be set to the first two slides of each event, and any two of critical items from each event would not appear one after the other (i.e., they would be interspersed with generic and control items).

To obtain a balanced design, two different versions of images were generated for each critical item by taking photos and editing them using Adobe Photoshop software (version 2017). Two different sets of critical items for each event were counterbalanced across participants. For example, in the misinformation group, one participant saw an image depicted that a man took a blue candy box and put it into the lady's bag, and later would read the misinformation of a red

candy box; whereas another participant saw an alternative image depicting that a man took a red candy box and put it into the lady's bag, and later would read the misinformation of a blue candy box (Supplementary Fig. 2). In total, there were 496 images (i.e., [50 + 12] × 8). All images were edited to a size of 1024 × 768 pixels.

For each event, 50 sentences were used to describe the content of the corresponding images as narratives (i.e., each sentence corresponds to each image). For critical items, their descriptions were accurate for the consistent group (e.g., the man took a blue candy box and put it into the lady's bag), vague for the neutral group (e.g., the man took a candy box and put it into the lady's bag [with the color of the box not specified]), and inaccurate for the misinformation group (e.g., the man took a red candy box and put it into the lady's bag). The misinformation descriptions presented details that were contradictory (not supplemental) to the original events. For control items, their descriptions were accurate for the consistent and misinformation groups (e.g., The lady was walking down the street, and there was a soda can next to her feet), and vague for the neutral group (e.g., The lady was walking down the street, and there was a piece of trash next to her feet). For generic items, their descriptions were accurate for all three groups. Each sentence consists of 13–18 Chinese characters. For each event, there are 24 questions pertaining to 12 critical items and 12 control items in the recognition test. Each question consists of 16–20 Chinese characters. Each option consists of 1–3 Chinese characters.

### Experimental procedure
There are three stages, including the original-event, post-event, and memory-test stages (Fig. 2a). Each of three stages includes eight events over four runs (i.e., two events per run), which takes about half an hour. There was a 3-h interval between original-event and post-event stages, and a 19-h interval between post-event and memory-test stages (i.e., a 23-h interval between original-event and memory-test stages). Participants were specifically instructed not to talk about these events with other people at any stage, and were asked to return to the lab individually as required. Exp. 1 was a behavioral study to compare the memory performance between three groups (i.e., misinformation, neutral, and consistent). All participants were shown eight events and given the memory test administrated on the computer using PsychoPy3 (version 3.2.4) in a quiet room. The only difference for participants in three groups was whether the post-event information included misleading, neutral, or consistent information for critical and control items. Exp. 2 was an fMRI study to explore the neural representations of false memory in the misinformation group, in which participants completed all three stages inside an fMRI scanner. Apart from this, the experimental materials and design were the same for these two experiments.

**Original-event stage.** In the first stage (i.e., original-event stage), participants saw images of eight events individually (i.e., 50 images per event). They were asked to remember each image of these events, and they would subsequently take a memory test about these events. Before the start of each event, there was a 3.5-s visual cue for the name of this event (e.g., "A man steals a girl's phone on the street"). Each trial started with a 0.5-s fixation point in the color of black. Next, an image was presented for 3.5 s. Each image was presented on the computer screen only once. To ensure that participants did not fall asleep in the fMRI scanner, for each event, a red fixation point was randomly displayed before two to four of the 26 generic items. Participants were asked to press a button as quickly and accurately as possible with the index finger of their right hand, when they saw the fixation point as shown in the color of red rather than black. At the end of each event, there was a 3.5-s visual cue indicating the end of this event (e.g., the end of "A man steals a girl's phone on the street"). The presentation order of these eight events and the two versions of critical items for each event was randomized across participants.

**Post-event stage.** In the second stage (i.e., post-event stage), participants were told that they would read narratives made by another eyewitness of these eight events (i.e., 50 sentences per event). Participants were unknowingly exposed to the post-event information, and not warned about potential discrepancies between images and narratives. Each sentence was presented on a horizontal line in the center of the screen. The presentation style of post-event information was the same as those in the original-event stage (e.g., start/end cues, fixation points, and stimulus presentation durations). To avoid the potential sequential effect between events, the presentation order of these 8 events was also randomized across participants (i.e., the event order of post-event was different from that of the original-event).

**Memory-test stage.** In the third stage (i.e., memory-test stage), participants were asked to answer questions in the recognition test, based on what they saw on the images of original events (i.e., 24 questions per event). Before the start of the memory test, all participants were given the same instructions (i.e., you saw the images and read the narratives, please try your best to answer the following questions based on what you saw on the images). For each question of critical items (e.g., What color of candy box did the man put into the lady's bag?), there are three options: (1) the detail presented on the image of original-event (e.g., blue), (2) the detail presented in the narrative of post-event for the misinformation group (e.g., red), and (3) a foil option (e.g., white) (Fig. 2b and Supplementary Table 3). For critical items, the endorsement rates of these three types of options were used as the indices of true memory, false memory, and foil, respectively. For each question of control items (e.g., When the lady was walking down the street, what was next to her feet?), there are three options: (1) the detail presented both in the image of original-event (e.g., soda can), (2) a foil option (e.g., banana peel), and (3) another foil option (e.g., plastic bag). For control items, the endorsement rate of the correct answer was used as the index of correct control, whereas that for both foils was used as the index of incorrect control.

Before the start of each event, there was a 9.5-s visual cue for the name of this event, in order to help participants to retrieve this event at the beginning. Before the start of each trial, there was a 0.5-s fixation point. Each trial lasted 9.5 s (Supplementary Fig. 3). A question was presented for 3.5 s. This design allowed participants to retrieve what they remembered before three options were presented. Then, three options were presented for 3 s. To ensure that participants gave due consideration to these three options, they were not allowed to respond during this period. After that, a black frame was presented around three options, indicating that participants were allowed to respond. They need to select one option by pressing 1 of 3 buttons with their index, middle, and ring fingers of right hand within 3 s. After their response, a red circle would be displayed around the number of option for their answer, so they could see that they provided a response. Three seconds after the black frame appears, the next trial would start. After completing all 24 questions for each event, there was a 3.5-s visual cue for ending this event. To obtain a balanced design for the memory test, the presentation order of these 8 events, the order of 24 questions within each event, and the on-screen locations (i.e., left, middle, and right) of three options for each question were randomized across participants. Participants were debriefed at the end of study. Additional information about the training prior to the formal experiment can be seen in the Supplementary Methods.

### Behavioral analysis
In the memory test in Exps. 1 and 2, the endorsement rates of original information (true memory), misinformation (false memory), foil, correct control, and incorrect control were calculated, respectively. In Exp. 1, the 3 by 5 mixed-effects ANOVA with post-hoc comparisons were used to explore the main effects and interaction between group type (i.e., misinformation, neutral, and consistent) and memory type

(i.e., true, false, foil, correct, and incorrect). For participants in the misinformation group of Exp. 1 or 2, the paired sample $t$ test was used to compare false memory with the endorsement rate of foils. To investigate whether the results of the misinformation group in Exp. 2 replicate those in Exp. 1, we used the 2 by 2 mixed-effects ANOVA to examine the interaction between group type (i.e., misinformation groups in Exp. 1 and 2) and memory type (i.e., false memory vs. the endorsement rate of foil). For the reaction time during the memory test in Exp. 1, the 3 by 4 mixed-effects ANOVA with post-hoc comparisons were used to explore the main effects and interaction between group type (i.e., misinformation, neutral, and consistent) and memory type (i.e., true, false, correct, and incorrect). The recognized foils were rare, so it was not included in the analysis of reaction time. For participants in Exp. 2, the paired sample $t$ test was used to explore the effect of memory type on the reaction time during the memory test. The packages of afex (version 1.1-1) and effectsize (version 0.7.0) in R (version 4.1.2) were used for ANOVA.

### fMRI data collection and analysis
**MRI acquisition.** Structural and functional imaging data were acquired on a 3.0T Siemens Prisma MRI scanner with a 64-channel head-neck coil at the MRI Center at Beijing Normal University. Using a simultaneous multi-slice EPI sequence, the high-resolution functional images were acquired (TR/TE/θ = 2000 ms/34 ms/90°; FOV = 200 mm × 200 mm; matrix = 100 × 100; in-plane resolution = 2 × 2 mm; slice thickness = 2 mm; multi-band acceleration factor = 3; slices = 72). Using a T1-weighted MPRAGE sequence, a high-resolution structural image was acquired for the whole brain (TR/TE/θ = 2530 ms/2.27 ms/ 7°; FOV = 256 mm × 256 mm; matrix = 256 × 256; slice thickness = 1 mm). A field map was acquired for correction of magnetic held distortions using a Gradient Echo sequence (TR = 750 ms; θ = 60°; TE1/TE2 = 5.20 ms/7.66 ms; FOV = 200 mm × 200 mm; matrix = 100 × 100; slice thickness = 2 mm; slices = 72).

**Image preprocessing.** Imaging data were preprocessed using fMRI-Prep v20.1.3[71]. Structural images were corrected for intensity non-uniformity with N4BiasFieldCorrection and were then skull-stripped by antsBrainExtraction.sh, using OASIS30ANTs as target template. Spatial normalization to the MNI152NLin2009cAsym template was performed through nonlinear registration by antsRegistration. For each functional run, the field map was co-registered to the reference volume and converted to a displacement map. The BOLD reference was coregistrated to the corresponding structural data by boundary-based registration with 6 degrees-of-freedom. Each functional run was slice-time corrected with 3dTshift (AFNI) and then resampled to their native space by applying a single, composited transform to correct for head-motion and susceptibility distortions. The BOLD time-series were resampled to the MNI152NLin2009cAsym template. Using the implementation of Nipype (version 1.15.1), the frame-wise displacement was calculated for each functional run. Then, data were filtered temporally with a nonlinear high-pass filter with a 100-s cutoff.

**Univariate analysis.** The General Linear Model (GLM; FSL's FILM module version 6.0.4) was used to model the data. We used smoothed data by a 5-mm full-width-half-maximum Gaussian kernel using FSL's SUSAN for the univariate analysis. During each one of the three stages, four types of trials were modeled: true memory, false memory, correct control, and incorrect control. The recognized foils and no-response trials were rare while generic items were not tested during the memory test, thus they were included as separate nuisance variables. In this model, the presentation of each trial (with a duration of 3.5 s for the original-event stage and the post-event stage and 9.5 s for the memory-test stage) was modeled as an impulse and convolved with a double gamma hemodynamic response function. The red fixation during the original-event and post-event stages were coded as an additional

nuisance variable. The GLM also included nuisance regressors for six motion parameters, framewise displacement (FD), and reaction time for all items during the memory test. We explored the univariate activations on each of the contrasts (true memory vs. false memory, true memory vs. correct control, or correct control vs. incorrect control) for the original-event, post-event, and memory-test stages, respectively. Each run was modeled in the first level analysis. Using a fixed-effects model, a higher-level analysis created cross-run contrasts for each participant. Using full FMRIB's Local Analysis of Mixed Effect 1 + 2 with automatic outlier detection, these contrasts were then input to a random-effects model for group analysis, with a height threshold of $Z > 2.3$ and a cluster probability of $p < 0.05$, corrected for whole-brain multiple comparisons using Gaussian Random Field Theory. Unless otherwise noted, the same threshold was used for univariate, neural pattern similarity, and correlational analyses. Brain images were visualized using Connectome Workbench (version 1.5.0) and BrainNet Viewer (version 1.7).

**Single-trial estimation.** The GLM (FILM version 6.00) was used to compute the $t$ map for each of the 400 trials during the original-event stage (3.5 s per trial), the 400 trials during the post-event stage (3.5 s per trial), and the 192 trials during the memory-test stage (9.5 s per trial), respectively. In this single-trial model, the presentation of each trial was modeled as an impulse and convolved with a double gamma hemodynamic response function[72]. Nuisance variables in each single-trial model were same as those used in the univariate analysis. To obtain reliable estimates of single-trial responses, the least squares separate method was used. The estimated $t$ value was obtained for each trial of each participant ($t$ value = beta value/square root of variance), and used to compute the neural pattern similarity in the following statistical analysis[73].

**Neural pattern similarity.** We calculated the neural pattern similarity between the original-event and post-event stages (OP), between the post-event and memory-test stages (PM), and between the original-event and memory-test stages (OM), for each of four memory types (i.e., true memory, false memory, correct control, and incorrect control), for the corresponding items and the non-corresponding items in the same event, separately (Fig. 3b). The recognized foils were rare while generic items were not tested during the memory test, thus they were not included. The item-specific representation was indicated by greater neural pattern similarity between each two of three stages for the corresponding items than that for the non-corresponding items in the same event.

The neural pattern similarity between each two of three stages for the corresponding items was calculated, by the Fisher's Z score reflecting neural pattern similarity of each item during one stage with its corresponding item during another stage. Take the OP for the corresponding items as an example, the pairwise correlation was calculated between the neural pattern of the narrative of one item during the post-event stage with the neural pattern of its corresponding image during the original-event stage. Then, these similarity scores were transformed into Fisher's Z scores, which were then averaged to generate the neural pattern similarity value for each type of trial. The same method was used to calculate for PM and OM, except that neural pattern similarities were computed for the corresponding item between original-event and memory-test stages and between post-event and memory-test stages, respectively.

As a baseline, the neural pattern similarity between each two of three stages for the non-corresponding items in the same event was calculated, by averaging the Fisher's Z scores reflecting neural pattern similarity of each item during the stage of post-event or memory-test, with the other 49 items in the same event during one of the previous stages. Take the OP for the non-corresponding items in the same event as an example, the pairwise correlations were calculated by averaging

the Fisher's Z scores reflecting neural pattern similarity of each narrative during the post-event stage with the other 49 images in the same event during the original-event stage. The same procedure was used for PM and OM, except that neural pattern similarities were computed for the tested item during the memory-test stage with the other 49 narratives during the post-event stage, and with the other 49 images during the original-event stage, respectively. In order to improve the signal-to-noise ratio, we used smoothed data by a 2-mm full-width-half-maximum Gaussian kernel for the neural pattern similarity analysis.

**Hippocampus as the region of interest.** To define anatomical region of interest (ROI) in each participant's native space, we segmented the structural image for each participant using FreeSurfer (version 6.0). Due to the key role of the hippocampus in false memory[13,20,22,65], we defined bilateral hippocampus based on its anatomical structure. The hippocampus was collapsed across both hemispheres. In the hippocampus, the neural pattern similarity (i.e., OP, OM, and PM for the corresponding items and the non-corresponding items) for each trial were calculated and then were averaged according to their memory type (i.e., true memory, false memory, correct control, and incorrect control). The 3 by 4 by 2 repeated measures ANOVA was used to explore the interaction and main effects of stage pairs (i.e., OP, OM, and PM), memory types (i.e., true memory, false memory, correct control, and incorrect control), and item specificity (i.e., corresponding and non-corresponding items) on the hippocampal pattern similarity. Next, we examined whether there was any simple two-way interaction between item specificity and stage pairs and between item specificity and memory types. Post-hoc $t$ test was used for further comparisons. In the hippocampus, the univariate activations (i.e., activations during the original-event, post-event, and memory-test stages) for each trial were extracted from the single-trial estimation, and then were averaged according to their memory type. The 3 by 4 repeated measures ANOVA was used to explore the interaction and main effects of stage types and memory types on the hippocampal activity levels.

**Individuation analysis for hippocampal representations.** To explore whether there was any participant-specific mapping between the neural (i.e., hippocampal item-specific representation of post-event information) and behavioral (i.e., false memory) data, we compared the within-participant and the averaged between-participant correlations. The logic here is that if a participant has a partially unique hippocampal item-specific representation of post-event information that produces false memory, then that participant's hippocampal item-specific representation of post-event information should predict this participant's own behavioral pattern of false memory better than other participants' false memory. To assess this possibility, we created a false-memory vector for each participant based on their specific behavioral pattern of false memory in the memory test. The false-memory vector in each participant was a binary vector of 192 values for the 192 tested items (i.e., 96 critical and 96 control items), with 1 indicating that the participant selected the misinformation option for the critical item (i.e., false memory), and 0 indicating that the participant selected the other options (i.e., true memory, foil, correct control, incorrect control, and no-response). To provide the within-participant correlation for each participant, we calculated the Spearman correlation between their hippocampal item-specific representations of post-event information and false-memory behavioral data. The correlation coefficients were transformed into Fisher's Z-scores. To provide the between-participant correlation, we then calculated the Spearman correlation between that participant's hippocampal item-specific representations of post-event information and false-memory data of each of the other participants, and averaged across these correlation values after Fisher's r-to-Z transformation. For each of 57 participants, this procedure resulted in a within-participant correlation and an

averaged between-participant correlation. Next, we used the paired $t$ test to compare the within- and between-participant correlations. To further ensure that this effect was specific for false memory, we performed an analysis similar to the one described above for correct control and true memory as comparisons.

**Correlating prefrontal activity with hippocampal item-specific representations.** We conducted the whole-brain voxel-wise analysis to examine whether prefrontal activity showed differential correlations with item-specific representations of original and post-event information in the hippocampus during the memory test, and whether this effect was greater for false memory than for correct control. Specifically, we examined the hippocampal-prefrontal correlation for the following contrast: (PM > OM: false memory > correct control) and (PM < OM: false memory > correct control). Because prefrontal activity might be associated with hippocampal activity rather than hippocampal representation, we used partial correlation analysis to control for the activation levels of the hippocampus during the memory test. The trial-by-trial partial correlation was calculated for false memory and correct control, separately. These coefficients were then transformed into Fisher's $Z$ scores for each participant, and used in the subsequent group analysis (an ordinary least square model with random effects).

Based on the above whole-brain analysis, we then defined two prefrontal ROIs (the left and right lateral prefrontal cortex) by including all the voxels in each cluster that showed suprathreshold values for the contrast (i.e., PM > OM: false memory > correct control). The Fisher's $Z$ scores were then extracted and averaged across voxels by condition and by ROI. We used the one-sample $t$-test to examine whether the averaged Fisher's $Z$ scores were significantly different from zero.

**Whole-brain searchlight analysis of cortical representations.** Using the searchlight method ($5 \times 5 \times 5$ cubes) by brainiak (version 0.10) in python 3 (version 3.7.9), we explored whether any other brain regions showed differences in item-specific neural pattern similarity between different memory types for OP, OM, and PM. We conducted a whole-brain searchlight analysis on each of the contrasts (true memory vs. false memory, true memory vs. correct control, and correct control vs. incorrect control) for OP, OM, and PM. To focus on item-specific representations, the results above were masked with the contrast of (corresponding > non-corresponding) for the appropriate memory type. For example, for the contrast of true memory > false memory, results were masked with the contrast of (corresponding > non-corresponding) for true memory. We used a random-effect model for group analysis. An ordinary least square model was used, since no first level variance was available.

**Individuation analysis for cortical representations.** We conducted the whole brain voxel-wise analysis to explore whether any other brain regions carried participant-specific representations of original or post-event information that predicted behavioral performance of true memory, false memory, or correct control.

**Informational connectivity between hippocampus and cortex.** We conducted the whole brain voxel-wise analysis to explore whether item-specific representations in the hippocampus and those in the cortex were positively correlated. Such a correlation would indicate informational connectivity[39,40], reflecting the functional connectivity between the hippocampus and the cortex. This analysis was based on the trial-by-trial covariation in information (item-specific representations of original or misinformation [i.e., OM or PM]) and was conducted separately for true and false memories. Spearman's correlation coefficients were used and then transformed into Fisher's $Z$ scores. We explored which brain regions had stronger informational connectivity with the hippocampus for true memory than false memory in OM, and which regions had stronger informational connectivity with the hippocampus for false memory than true memory in PM. For this exploratory analysis, we used a relatively liberal threshold to find these regions ($Z > 1.7$ and a cluster probability of $p < 0.05$, corrected for whole-brain multiple comparison using Gaussian random filed theory). Finally, the above results were masked to ensure that correlation coefficients were higher than zero for true memory in OM, and were higher than zero for false memory in PM. For example, for the contrast of true memory minus false memory in OM, results were masked with brain regions showing a positive correlation with the hippocampus for true memory in OM.

### Reporting summary

Further information on research design is available in the Nature Portfolio Reporting Summary linked to this article.

## Data availability

The behavioral and fMRI data generated in this study have been deposited on OpenNeuro [https://openneuro.org/datasets/ds004261]. The data illustrated in the figures are provided in the Source Data file. Source data are provided with this paper.

## Code availability

The code in this study has been deposited on OpenNeuro [https://openneuro.org/datasets/ds004261].

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

## Acknowledgements

This study was supported by the National Natural Science Foundation of China 31971000 to B.Z. and Young Top Notch Talents of Ten Thousand Talent Program to B.Z.. We thank Y. Liang, X. Chen, and M. Guo for their assistance with preparation of experimental materials and data collection.

## Author contributions

Conceptualization by B.Z.; methodology by B.Z., X.S., and A.L.; investigation by X.S. and A.L.; analysis by B.Z., X.S., and A.L.; software by X.S. and A.L.; writing by B.Z., X.S., C.C., and E.F.L.; funding acquisition by B.Z.; supervision by B.Z., C.C., and E.F.L.

## Competing interests

The authors declare no competing interests.
