## [Peer Review File · Nature Communications]

Cross-stage neural pattern similarity in the hippocampus predicts false memory derived from post-event inaccurate informationReviewer #1 (Remarks to the Author):

The manuscript assessed three different theoretical theories that have been proposed to explain the misinformation effect. To do so the authors used fMRI and representational similarity analysis (RSA) that focussed on similarity between different stages of misinformation-induced false memory (i.e., original event, post-event misinformation, and memory test). Although these models (i.e., the non-retention theory, the trace-alteration theory, and the multiple-traced theory) posit different pattern of changes in hippocampal representations, it is largely unknown how these representations change across stages to trigger false memory. To investigate the dynamic changes of representational content across stages, the authors used RSA ROI analysis within the hippocampus (in the main paper), a searchlight analysis (reported in supplemental material), an individuation analysis, and univariate analysis (reported in supplemental material). The findings support the multiple trace memory theory and suggest that the prefrontal cortex is involved in source misattribution by modulating the item-specific representations in the hippocampus.

Although the topic is timely and the theoretical justification for the research is appropriate, the experimental design and the subsequent results are subjected to confounds. Thus, given the interpretability issue, it is problematic, if not impossible, to disentangle whether the current manuscript can make a potential contribution to the field of misinformation-induced false memory.

In particular, my major concern is about the interpretation of the reinstatement of both the original event and the misinformation during the test phase. In this study, the subjects viewed a series of images for each event during the original phase (e.g., viewing "The man took a red candy box and put it into the lady's bag"). Then, during the post-event phase, participants read the misinformation even about the colour of the candy box (e.g., "The man took a blue candy box and put it into the lady's bag"). During the test phase, a three-alternative force-choice test was used asking ("What colour of candy box did the man put into the lady's box"). The problem is that both the post-event phase and the memory test used words that share the same meaning with the original presentation of the scenes. Thus, the observed findings in phase 2 (i.e., post-event) or 3 (i.e., memory test) may have been related to memory reactivation or to the reprocessing of scenes elicited by using words. This is a serious issue for this manuscript since it is difficult to understand whether the observed activity pattern between stages is due reinstatement of original information/misinformation (i.e., a real memory effect) or due to reprocessing of the scene (i.e., perception). To avoid this confound, typical studies that want to isolate neural patterns related to memory reactivation, employ cued recall tests in which only associates are used as cue during retrieval and the critical information is not presented again.

Another problem for the interpretation of the results is that for true memory, the original trace is retained in the post event phase, but not in the test phase, while for false recognition the original trace is retained in both the post event phase and in the memory test. According to the multiple-trace theory, for true recognition I should also be able to see similarity between the original event and the memory test, a reinstatement that would allow participants to correctly remember that, e.g., "the candy box was blue". But this is not the case. This may suggest that, in some cases, although the original event is re-represented in part during the post event, not necessarily survives during the test phase. This may explain the low endorsement rate (0.4, similar to false memory) for studied information. Results do not seem to be that strong in favour of one or the other theory, although correlation between OM and PM for false memory may seem to suggest otherwise.

Reviewer #2 (Remarks to the Author):

In the submitted paper for consideration in Nature Communications, Shao et al., aimed

to identify the neural correlates associated with misinformation and its effects on later memory. In this study, across two experiments, Shao et al., attempted to dissociate between various theoretical perspectives on misinformation using functional neuroimaging: non-retention, trace-alteration, and multiple-trace memory models. In Experiment 1, Shao et al., report a behavioral study employing a standard misinformation paradigm (e.g., Okado & Stark, 2005), consisting of three stages: original-event, post-event, memory-stage. There were three groups of participants: misinformation, neutral, consistent information group. The authors reported a significant misinformation effect: the misinformation group led to more false memory than did the neutral and consistent groups. In Experiment 2, functional neuroimaging was employed to identify the neural correlates of the misinformation effect by conducting a representational similarity analysis and comparing hippocampal pattern similarity across three stages as a function of the memory response types (i.e., true memory, false memory, foil, correct control, and incorrect control). Results showed that in the memory test stage, hippocampal reinstatement of original information faded for true memory, whereas hippocampal reinstatement of post-event misinformation competed with original information to create false memory. A secondary analysis showed that regions of the medial prefrontal cortex showed a negative correlation with the hippocampal item-specific reinstatement of original information when false memory occurred during the memory test. Taken together, these findings are interpreted as support for multiple-trace memory theory, which posits that two opposing memory traces of the original and misleading information coexist in the hippocampus and compete with each other in leading to false memory.

The manuscript covers a timely topic concerning the neural correlates associated with misinformation and false memory. I applaud the authors in employing an innovative analytical approach together with a well-validated behavioral paradigm to induce false memory. My comments, which are detailed below, concern largely the analytical approach which I believe do not support the authors conclusion. In addition, greater clarity is needed in describing the analysis, as well as more data for each condition (i.e., data from the corresponding and non-corresponding conditions).

1. The authors highlight the role of the medial prefrontal cortex in resolving conflict. There is a clear lack of rationale for this statement. What is the evidence for this? I was surprised the authors did not include the inferior frontal gyrus given its long-standing role in resolving interference (e.g., Badre & Wagner, 2007, Carpenter et al., 2021). As written it seems as if the authors ran a searchlight and identified effects in the medial prefrontal cortex and then included it in the Introduction (i.e., post-hoc hypothesis generation).

Similarly, in the Results section, it is stated that "the prefrontal cortex is expected to be involved in source misattribution". This is a highly simplified perspective. The prefrontal cortex is not a homogenous region. It is the dorsal, not medial, regions of the prefrontal cortex that have been implicated in evaluating/monitoring processes during memory (e.g., Dobbins et al., 2003, *Neuropsychologia*; Mitchell, et al., 2004, *JOCN*).

2. After reading the introduction I was left unconvinced at the justification and motivation for 'randomization' as a way to target hippocampal processing. A rather large assumption is made that through such a manipulation participant's 'must rely on the hippocampus to navigate and retrieve the corresponding information presented in the previous stages.'

3. It would be of great interest to conduct a searchlight analysis within the hippocampus to identify the regions that show differential effects of stage pair as a function of memory condition. As of now, it is unknown which hippocampal voxels/regions are contributing to the differential pattern similarity. Such an analysis would also answer questions regarding possible anterior-posterior hippocampal dissociations that are now well-recognized in the field (e.g., Poppenk et al., 2013, *TICS*).

4. There are certain sections of the paper that need more analytical detail. For example, how was the prefrontal-to-hippocampal correlation analysis conducted and how was it statistically thresholded?

5. There are well-known limitations of a searchlight analysis that should be acknowledged (see, Etzel et al., 2013, *Neuroimage*). I was very surprised that the hippocampal effects were not replicated in the searchlight analysis that identified the

medial prefrontal effects. Some discussion of this issue is warranted.

6. The term, neural activity pattern similarity, is very confusing and should be replaced.

7. In the example on page 10, is the greater similarity due to the encoding of the novel information of 'red candy box' during the post-event phase or retrieving 'blue candy box' from the prior original-event phase? Why would reactivating "the man took a blue candy box" from the original event during the presentation of "the man took a red candy box" during the post-event stage lead to greater true memory? One would assume that such item-specific reactivation would lead to false memory due to the blurring/similarity of the two pieces of encoded information? More clarity is needed on the mechanisms at play here.

8. I was surprised by the lack of an effect for the OM condition during true memory? There should be evidence of reinstatement for the correct detail (i.e., original event information) when recollecting it accurately (i.e., during the memory stage)? In contrast, it was found that OP correlated with accurate/true memory, which is surprising.

9. The pattern of results for the false memory condition seems to contradict the predicted pattern of results. All three stage pairs do not differ with respect to pattern similarity (i.e., OP, PM, and OM do not statistically differ). Presumably, there should be a selective increase in the PM condition as the post-event information is being reinstated over and above the original information leading to false memory.

10. Although there was a significant 3-way interaction on the pattern similarity metrics, the authors fail to conduct and report the subsidiary ANOVA's to determine which pairs of memory conditions are driving the 3-way interaction. For example, based on the follow-up tests reported, the false memory and incorrect control conditions would not yield a significant interaction and thus should be collapsed, yet the authors compare the stage pairs across each memory condition. In addition, please report all parent ANOVA results (not simply the 3-way interaction). I do not believe that the current statistical approach allows the authors to claim specificity with respect to memory condition and stage pair. For example, the authors claim that: 'The analyses reported above suggested that there were two memory traces (i.e., significant OM and PM values) in the hippocampus for false memory and correct controls during the memory test'. However, they do not directly compare the data with an ANOVA for these two memory conditions. The same applies for the following statement, 'item-specific reinstatement of post-event information (i.e., PM) in the hippocampus was greater for false memory and correct control than for true memory.' The authors need to directly compare the data from these conditions with respective ANOVA's (e.g., collapsing the false memory and correct control (assuming a null interaction), and then compared to the true memory data).

11. It is necessary to provide the pattern similarity data separately for the corresponding and non-corresponding trials. The authors only provide the difference data. It is unknown whether the greater than zero item-specific reinstatement is due to a decrease from zero for the non-corresponding trials or an increase for corresponding trials. Obviously, it should be the former to support the logic of the analysis.

12. The individuation analysis is quite interesting. One would like to see a similar analysis done to predict true memory. Such a finding would support models of hippocampal function during episodic memory (i.e., reinstatement of item-specific information supports veridical recollection; Chadwick et al., 2010, Neuron; Tomparry et al., 2016, Hippocampus; Norman & O'Reilly, 2003; Xue, 2018, TICS).

13. As a whole greater clarity is needed across the different conditions and what comparisons support what predictions/patterns across the three theoretical models.

14. Please add bars denoting which stage pairs (OP, OM, PM) statistically differ for each memory condition (true, false, correct control, and incorrect control).

Reviewer #3 (Remarks to the Author):

The manuscript by Shao and colleagues describes a pair of human experiments (behavioral and fMRI) investigating the misinformation effect. The experiments use a standard experimental structure where events are first encoded (original memories),

then subjects are subsequently exposed (in a post-event stage) to information that is consistent, neutral, or inconsistent (misinformation) with the original memory and finally memory for the original event is tested. As is standard, they report a behavioral effect where subjects are more likely to endorse false information when they have been exposed to that (mis)information during the post-encoding phase. While the behavioral effects are simply a replication of a well-established finding, the experimental materials used here are quite nice in that the events were designed to have a lot of individual elements which could then be cued and tested—something that is important for conducting an adequately powered fMRI study. For the fMRI experiment, the main question is whether the tendency to mistakenly 'remember' the misinformation (i.e., false memories) is related to hippocampal pattern similarity across the three memory stages: the original event, the post-event stage, and the memory test. The key and most interesting finding is that true vs. false memories are associated with different patterns of hippocampal pattern similarity across stages. In relative terms, similarity between the original event and the post-event stage is greater for true than false memories whereas similarity between the post-event stage and the final memory test is greater for false than true memories. This pattern of results is consistent with an interpretation wherein reinstatement of the original event during the post-event stage helps prevent falsely remembering the misinformation whereas reinstatement of the misinformation during the final memory test has the opposite effect (i.e., it leads to false memories). A second analysis shows that there is a subject-specific 'alignment' of the fMRI pattern similarity results with the specific elements that are behaviorally mis-remembered. Finally, it is shown that prefrontal activity is negatively correlated with the reinstatement of the original memory during the final test, which is interpreted as reflecting inhibition of the original memory.

There are a number of appealing elements to this paper and it is generally well written and clearly presented. In particular, it is a nice experimental design with a robust sample size and with a clear presentation of competing theoretical accounts and how the current experiment can test these. The first two findings are very nice and I think they will be of relatively broad interest. The last finding (related to 'inhibition' of true memories) was less compelling to me. The main weaknesses in the paper are (1) while I do think the current paper represents an advance, there are, as the authors note, several prior papers which have addressed conceptually similar ideas—though, with less sophisticated approaches and (2) I think the paper gets into a bit of trouble by tending to present interpretations/assumptions instead of results and by failing to consider alternative accounts. Taken together, while my overall impression is somewhat favorable, there are some notable weaknesses in the current form of the manuscript.

Comments.

1. I believe the biggest issue in the manuscript is over-interpretation of the findings. There are three main examples of this.

a. First, fMRI pattern similarity is taken as evidence of reactivation without any consideration of alternative explanations. However, item-specific similarity could also easily be explained by the fact that within-item comparisons tend to have a lot of shared semantic information. So, even if there was no memory component whatsoever, I would expect some degree of similarity (greater for within than across items) just because of shared semantic information. To be fair, I would not really expect a "semantic only" effect in the hippocampus, but the point is that the similarity measure is not a pure test of episodic memory because it is confounded with semantic similarity. As such, rather than describing the results as "reinstatement," it would be more appropriate to describe the results more neutrally as "pattern similarity" (since this is what is actually measured). Obviously, higher pattern similarity might reflect reinstatement, but that is an interpretation. To be clear, I do think that the most likely explanation of the pattern similarity effects is a "reinstatement account"—but I just think that there should be a distinction between the result and the interpretation.

b. Similarly, it is argued (e.g., Line 231) that the pattern similarity analyses suggest “that there were two memory traces (i.e., significant OM and PM values).” And elsewhere in the paper it is more or less taken as fact that there were “competing” memory traces. But, again, this is an interpretation. From my perspective, I did not find the evidence of competing memory traces to be compelling. In particular, the PM value could, in principle, reflect reinstatement of the exact same information that is reflected in the OP measure. In fact, one way of looking at the data is that PM similarity does not exceed OM similarity for any of the conditions. In other words, the representation at the memory test is no more similar to the post-event stage than it is to the original event. Obviously, the fact that PM similarity is greater for false memories fits nicely with the interpretation that there is a competing trace (of the misinformation), but, again, that is an interpretation. An alternative interpretation, for example, could be that if the ‘wrong’ information from the original event is reinstated at the post-event stage and then again at the final memory test (e.g., remembering a feature of the original event that is not related to the misinformation), it results in a false memory. I am not saying I find this account more likely, but the key point is that I think there is a plausible account where the relatively high OM/PM similarity in the false memory condition reflects reinstatement of SOMETHING from the original memory as opposed to reinstatement of the misinformation. To make a stronger claim about distinct memory traces, I think it would be critical to statistically test whether the post-event representation explains unique variance in the memory-test representation (that is, variance that is not explained by the original event representation).

c. Finally, the idea that prefrontal cortex inhibited the original memory trace feels like a stretch to me. Frankly, I really do not know what to make of the prefrontal data. The argument about inhibition feels a bit backwards to me. Namely, it seems more intuitive that the prefrontal cortex would play a role in inhibiting the (incorrect) post-event memory as opposed to the original memory. Thus, I found the prefrontal correlation results to be somewhat puzzling and did not find the inhibition interpretation intuitive or compelling.

2. For the prefrontal correlation analyses, it is reported that similar effects were not observed for the control condition, but a stronger test would be to actually test whether effects were statistically greater for the false memory than control condition. But this would have to be done within the searchlight analysis.

3. For the second main question (starting on Line 230), the question is framed as whether “the results reflected reinstatements of post-event information shared among all research participants, or there might be consistent individual differences.” This framing was quite confusing to me. Ultimately, I understand the analysis and I liked the result, but I think this could be framed in a more effective way.

4. In initially reading the paper, I was a bit surprised by the lack of consideration of any region other than the hippocampus. This is only addressed at the very end of the Results where the reader is oriented to the supplement. I do appreciate that the supplement reports whole-brain analyses. However, it’s a bit hard to get a full sense of how the effects in the hippocampus compare to those in other regions. From what I can tell, it does seem that the pattern of results in many cortical areas is qualitatively dissimilar to the hippocampus. While that is not problematic in and of itself, it does complicate some of the conclusions. That is, many of the conclusions in the paper are framed as “general” conclusions about the basis of the misinformation effect; but, really, they are specific to the hippocampus. For example, it appears from Figure S7 that quite a few regions show stronger OM similarity for true vs. false memories. In fact, OM similarity looks like it may be a more robust predictor of true memories (in terms of number of regions) than the other similarity measures. This contrasts with the data from the hippocampus and, in fact, may suggest a mechanistically distinct way in which false memories are avoided. While I appreciate that the authors have an a priori interest in the hippocampus—and I think that makes sense—I do think that some conclusions in the paper need to be more

carefully presented given that the pattern of data from the hippocampus appears to qualitatively differ from many other brain regions.

Reply to Reviewer 1's comments:

The manuscript assessed three different theoretical theories that have been proposed to explain the misinformation effect. To do so the authors used fMRI and representational similarity analysis (RSA) that focussed on similarity between different stages of misinformation-induced false memory (i.e., original event, post-event misinformation, and memory test). Although these models (i.e., the non-retention theory, the trace-alteration theory, and the multiple-traced theory) posit different pattern of changes in hippocampal representations, it is largely unknown how these representations change across stages to trigger false memory. To investigate the dynamic changes of representational content across stages, the authors used RSA ROI analysis within the hippocampus (in the main paper), a searchlight analysis (reported in supplemental material), an individuation analysis, and univariate analysis (reported in supplemental material). The findings support the multiple trace memory theory and suggest that the prefrontal cortex is involved in source misattribution by modulating the item-specific representations in the hippocampus.

Although the topic is timely and the theoretical justification for the research is appropriate, the experimental design and the subsequent results are subjected to confounds. Thus, given the interpretability issue, it is problematic, if not impossible, to disentangle whether the current manuscript can make a potential contribution to the field of misinformation-induced false memory.

1.1. In particular, my major concern is about the interpretation of the reinstatement of both the original event and the misinformation during the test phase. In this study, the subjects viewed a series of images for each event during the original phase (e.g., viewing “The man took a red candy box and put it into the lady’s bag”). Then, during the post-event phase, participants read the misinformation even about the colour of the candy box (e.g., “The man took a blue candy box and put it into the lady’s bag”). During the test phase, a three-alternative force-choice test was used asking (“What colour of candy box did the man put into the lady’s box”). The problem is that both the post-event phase and the memory test used words that share the same meaning with the original presentation of the scenes. Thus, the observed findings in phase 2 (i.e., post-event) or 3 (i.e., memory test) may have been related to memory reactivation or to the reprocessing of scenes elicited by using words. This is a serious issue for this manuscript since it is difficult to understand whether the observed activity pattern between stages is due reinstatement of original information/misinformation (i.e., a real memory effect) or due to reprocessing of the scene (i.e., perception). To avoid this confound, typical studies that want to isolate neural patterns related to memory reactivation, employ cued recall tests in which only associates are used as cue during retrieval and the critical information is not presented again.

Response:

We thank Reviewer 1 for pointing out this issue. We acknowledge that there are perceptual overlaps between stages in our experiment. However, based on the following evidence, we argue that cross-stage neural pattern similarity is primarily caused by memory representations of specific details rather than by perceptual reprocessing of the scene.

First, there was a significant effect of memory type on the hippocampal pattern similarity between post-event and memory-test stages. Specifically, in the hippocampus, the item-specific neural pattern similarity between post-event and memory-test stages was higher for false memory than for true memory (Figure 3d in the manuscript). If the hippocampal pattern similarity were purely a reflection of perceptual reprocessing across stages, there should be no such a difference, because for all memory types, participants underwent perceptual reprocessing of scenes across stages and saw all three options for each question in the memory test.

Second, the behavioral performance of false memory, but not that of correct control, was predicted by the hippocampal pattern similarity between post-event and memory-test stages (Figure 4 in the manuscript). If the hippocampal pattern similarity were purely a reflection of perceptual reprocessing across stages elicited by words, there should be no such a correlation for false memory, because both critical and control items involved the same kind of perceptual reprocessing in the memory test. In other words, our findings suggest that the pattern similarity between post-event and memory-test stages in the hippocampus was primarily caused by memory representations.

We agree with this reviewer that future studies should use cued recall tests to avoid potentially confounding effects of perceptual overlap between stages. Therefore, we added the following sentences to the discussion.

Discussion, pages 29~30, lines 497~512:

“Another limitation of the current study was that we did not use cued recall tests to isolate neural patterns related to memory reactivation. In this study, the recognition test allowed for better control of the stimuli, but also enabled the reprocessing of scenes elicited by words in the memory test. However, we do not believe that perceptual reprocessing can explain our results because (a) item-specific neural pattern similarity between post-event and memory-test stages was higher for false memory than for true memory, and (b) this neural pattern similarity was associated with the behavioral performance of false memory rather than correct control. If perceptual reprocessing had

been responsible for the neural pattern similarity, we would not have seen these results because both critical and control items involved the same kind of perceptual reprocessing in the memory test. Therefore, we believe that cross-stage neural pattern similarity was primarily caused by memory representations of specific details rather than by perceptual reprocessing of scenes. Nevertheless, future studies should consider using cued recall tests to avoid any potential confounding of perceptual reprocessing and to improve the ecological validity of memory studies (Cohn-Sheehy et al., 2021; Maguire, 2022).”

1.2. Another problem for the interpretation of the results is that for true memory, the original trace is retained in the post event phase, but not in the test phase, while for false recognition the original trace is retained in both the post event phase and in the memory test. According to the multiple-trace theory, for true recognition I should also be able to see similarity between the original event and the memory test, a reinstatement that would allow participants to correctly remember that, e.g., “the candy box was blue”. But this is not the case. This may suggest that, in some cases, although the original event is re-represented in part during the post event, not necessarily survives during the test phase. This may explain the low endorsement rate (0.4, similar to false memory) for studied information. Results do not seem to be that strong in favour of one or the other theory, although correlation between OM and PM for false memory may seem to suggest otherwise.

Response:

Based on kind suggestions of reviewers, we have conducted additional analysis of true memory and added the interpretation of results in this revision. Regarding the neural basis of true memory, our results showed that the hippocampus no longer carried the original information during the memory test, instead it was carried by the lateral parietal cortex (as evidenced by item-specific neural pattern similarity between original-event and memory-test stages related to behavioral performance of true memory in the misinformation group [see Figure 6b in the manuscript]). These findings are consistent with previous studies showing that the involvement of the lateral parietal cortex was sustained, but the involvement of the hippocampus was transient in order to produce true memory during the memory test (Staresina & Wimber, 2019; Vilberg & Rugg, 2012).

As for the reviewer's concern about the "low" level of true memory in the misinformation group, we believe that it was actually not low and was comparable to previous studies. Specifically, the endorsement rates of true memory (about 0.49 and 0.44 in Exp. 1 and 2) were higher than that of foil (about 0.09 and 0.14 in Exp. 1 and 2), suggesting significant true memory. More importantly, they are within the range of the rates found in previous studies, from 0.19 to 0.62 (Karanian et al., 2020; Loftus, 2005; Loftus et al., 1978; Okado & Stark, 2005; Putnam et al., 2017; Stark et al., 2010; Zhu et al., 2016). Behavioral performance of true memory depends on various factors, including time interval, task difficulty, and experimental manipulations (Loftus, 2005). Our rates were closest to those of Karanian et al. (2020), who reported that the mean values of true memory in their two experiments were 0.42 and 0.37 under the condition of misinformation without warning.

We adjusted our manuscript to integrate the above discussion, but because the changes involve many sections of the revised manuscript (pages 21~23, 27~29 and Figure 6), we do not post all edits here. With this clarification of our results on true memory, we believe that our results favor the multiple-trace model of false memory over the other two.

Reply to Reviewer 2's comments:

In the submitted paper for consideration in Nature Communications, Shao et al., aimed to identify the neural correlates associated with misinformation and its effects on later memory. In this study, across two experiments, Shao et al., attempted to dissociate between various theoretical perspectives on misinformation using functional neuroimaging: non-retention, trace-alteration, and multiple-trace memory models. In Experiment 1, Shao et al., report a behavioral study employing a standard misinformation paradigm (e.g., Okado & Stark, 2005), consisting of three stages: original-event, post-event, memory-stage. There were three groups of participants: misinformation, neutral, consistent information group. The authors reported a significant misinformation effect: the misinformation group led to more false memory than did the neutral and consistent groups. In Experiment 2, functional neuroimaging was employed to identify the neural correlates of the misinformation effect by conducting a representational similarity analysis and comparing hippocampal pattern similarity across three stages as a function of the memory response types (i.e., true memory, false memory, foil, correct control, and incorrect control). Results showed that in the memory test stage, hippocampal reinstatement of original information faded for true memory, whereas hippocampal reinstatement of post-event misinformation competed with original information to create false memory. A secondary analysis showed that regions of the medial prefrontal cortex showed a negative correlation with the hippocampal item-specific reinstatement of original information when false memory occurred during the memory test. Taken together, these findings are interpreted as support for multiple-trace memory theory, which posits that two opposing memory traces of the original and misleading information coexist in the hippocampus and compete with each other in leading to false memory.

The manuscript covers a timely topic concerning the neural correlates associated with misinformation and false memory. I applaud the authors in employing an innovative analytical approach together with a well-validated behavioral paradigm to induce false memory. My comments, which are detailed below, concern largely the analytical approach which I believe do not support the authors conclusion. In addition, greater clarity is needed in describing the analysis, as well as more data for each condition (i.e., data from the corresponding and non-corresponding conditions).

2.1. The authors highlight the role of the medial prefrontal cortex in resolving conflict. There is a clear lack of rationale for this statement. What is the evidence for this? I was surprised the authors did not include the inferior frontal gyrus given its long-standing role in resolving interference (e.g., Badre & Wagner, 2007, Carpenter et al., 2021). As written it seems as if the authors ran a searchlight and identified effects in the medial prefrontal cortex and then included it in the Introduction (i.e., post-hoc hypothesis generation). Similarly, in the Results section, it is stated that “the prefrontal cortex is expected to be involved in source misattribution”. This is a highly simplified perspective. The prefrontal cortex is not a homogenous region. It is the dorsal, not medial, regions of the prefrontal cortex that have been implicated in evaluating/monitoring processes during memory (e.g., Dobbins et al., 2003, Neuropsychologia; Mitchell, et al., 2004, JOCN).

Response:

We apologize for the confusion. In the original manuscript, we only stressed the role of medial prefrontal cortex in resolving interference, given its function of suppressing inappropriate memory traces in the hippocampus (Eichenbaum, 2017; Preston & Eichenbaum, 2013). However, we agree with this reviewer that the prefrontal cortex is a heterogeneous region. When confronted with the multiple traces in the hippocampus, different parts of the prefrontal cortex might be involved. Based on kind suggestions of this reviewer, we have clarified the rationale for the hippocampal-prefrontal correlation analysis and added these references in the revised manuscript.

Introduction, page 5, lines 82~90:

“A key mechanism involved here is cognitive control, which should be subserved by certain parts of the prefrontal cortex (Badre & Wagner, 2004; Simons & Spiers, 2003). To resolve the competition between two memory traces in the hippocampus during the memory test, the ventrolateral, medial, and dorsolateral prefrontal cortex may be involved in selecting goal-relevant memory of original information (Badre & Wagner, 2007; Carpenter et al., 2021), suppressing inappropriate memory of misinformation (Eichenbaum, 2017; Preston & Eichenbaum, 2013), and monitoring the discrepancy between misinformation and original information (Dobbins et al., 2003; Mitchell et al., 2004). However, it was unclear which parts of the prefrontal cortex might work in concert with the hippocampus to resolve this conflict during the memory test.”

We also added the relevant information to the method, results, and discussion (pages 18~20, 26~27, 47). Because they involve many sections of the manuscript, the revised text are not copied here, but some of the text that were specific to related comments from Reviewers 2 and 3 are included in our response to those comments in the following pages.

2.2. After reading the introduction I was left unconvinced at the justification and motivation for 'randomization' as a way to target hippocampal processing. A rather large assumption is made that through such a manipulation participant's 'must rely on the hippocampus to navigate and retrieve the corresponding information presented in the previous stages.

Response:

We apologize for making the assumption about randomization and hippocampal processing. Indeed, the well-recognized role of randomization in functional imaging data is to avoid the confounding of temporal drift (Henriksson et al., 2015). In this revision, we have deleted the original sentence and replaced it with the following sentence.

Introduction, page 7, lines 115~116:

"This randomization helped to avoid potential confounding effects of temporal drift and sequence-related structure on the neural pattern similarity estimation (Henriksson et al., 2015)."

2.3. It would be of great interest to conduct a searchlight analysis within the hippocampus to identify the regions that show differential effects of stage pair as a function of memory condition. As of now, it is unknown which hippocampal voxels/regions are contributing to the differential pattern similarity. Such an analysis would also answer questions regarding possible anterior-posterior hippocampal dissociations that are now well-recognized in the field (e.g., Poppenk et al., 2013, TICS).

Response:

Following this insightful suggestion, we conducted a searchlight analysis within the hippocampus to identify the regions that showed differential effects of stage pairs as a function of memory type. However, we did not find such a region. One possible reason is that the searchlight method has several limitations, including mismatches in size and shape of relevant brain regions and those of the searchlight (Etzel et al., 2013).

To answer the question regarding possible anterior and posterior hippocampal dissociations (Poppenk et al., 2013), we defined the anterior and posterior hippocampus as two regions of interest (i.e., based on the location of uncal apex in native space), and then conducted the repeated-measure ANOVA on the data. We added the results of cross-stage neural pattern similarity in the anterior and posterior hippocampus in the revised manuscript and Supplementary Table 8 in the Supplementary Information.

Results, page 15, lines 235~239:

"Additional results regarding the anterior and posterior hippocampus indicated a functional dissociation along the long axis of the hippocampus for true memory and correct control but not for false memory and incorrect control (Details are shown in Supplementary Table 8)."

Supplementary Table 8. The mean and standard deviation (Mean \pm SD) for the anterior and posterior hippocampal pattern similarity across the three stages for corresponding items, non-corresponding items in the same event, and item-specific representation for each memory type in Exp. 2.

	Critical		Control	
	True	False	Correct	Incorrect
Anterior hippocampus				
OP				
corresponding	0.0061 \pm 0.0792	0.0042 \pm 0.0784	0.0048 \pm 0.0788	0.0033 \pm 0.0805
non-corresponding	0.0014 \pm 0.0305	0.0021 \pm 0.0302	0.0018 \pm 0.0312	0.0014 \pm 0.0303
item-specific	0.0047 \pm 0.0759	0.0021 \pm 0.0745	0.0030 \pm 0.0745	0.0019 \pm 0.0750
OM				
corresponding	0.0005 \pm 0.0799	0.0010 \pm 0.0797	0.0024 \pm 0.0793	0.0034 \pm 0.0811
non-corresponding	0.0008 \pm 0.0310	0.0013 \pm 0.0312	0.0006 \pm 0.0313	0.0016 \pm 0.0306
item-specific	-0.0003 \pm 0.0751	-0.0003 \pm 0.0745	0.0018 \pm 0.0749	0.0018 \pm 0.0768
PM				
corresponding	0.0035 \pm 0.0789	0.0053 \pm 0.0803	0.0041 \pm 0.0798	-0.0006 \pm 0.0808
non-corresponding	0.0017 \pm 0.0296	0.0019 \pm 0.0304	0.0019 \pm 0.0306	0.0016 \pm 0.0308
item-specific	0.0018 \pm 0.0743	0.0034 \pm 0.0740	0.0022 \pm 0.0754	-0.0022 \pm 0.0774
Posterior hippocampus				
OP				
corresponding	0.0002 \pm 0.0822	0.0008 \pm 0.0806	-0.0014 \pm 0.0807	0.0004 \pm 0.0798
non-corresponding	-0.0009 \pm 0.0311	-0.0004 \pm 0.0310	-0.00003 \pm 0.0313	-0.0004 \pm 0.0304
item-specific	0.0011 \pm 0.0778	0.0012 \pm 0.0759	-0.0014 \pm 0.0771	0.0008 \pm 0.0762
OM				
corresponding	0.0002 \pm 0.0836	0.0036 \pm 0.0823	0.0041 \pm 0.0824	-0.0003 \pm 0.0818
non-corresponding	0.0014 \pm 0.0317	0.0015 \pm 0.0309	0.0009 \pm 0.0309	0.0016 \pm 0.0310
item-specific	-0.0012 \pm 0.0792	0.0021 \pm 0.0774	0.0032 \pm 0.0777	-0.0019 \pm 0.0768
PM				
corresponding	-0.0030 \pm 0.0835	0.0008 \pm 0.0817	0.0031 \pm 0.0823	0.0022 \pm 0.0818
non-corresponding	-0.0010 \pm 0.0308	-0.0005 \pm 0.0310	-0.0012 \pm 0.0307	0.0005 \pm 0.0306
item-specific	-0.0020 \pm 0.0786	0.0013 \pm 0.0767	0.0043 \pm 0.0783	0.0017 \pm 0.0779

Note. OP: Neural pattern similarity between original-event and post-event stages. OM: Neural pattern similarity between original-event and memory-test stages. PM: Neural pattern similarity between post-event and memory-test stages. Bolded scores represent item-specific neural pattern similarity, which was calculated by the neural pattern similarity for corresponding items minus that for non-corresponding items in the same event for each type of memory. True: true memory of critical items. False: false memory of critical items. Correct: correct control items. Incorrect: incorrect control items. For each participant, the whole hippocampus was divided into its anterior and posterior segments based on the location of uncus apex in the native space.

“Results of anterior and posterior hippocampal pattern similarity

The 3 (stage pairs) × 4 (memory types) × 2 (item specificity) × 2 (ROI: anterior and posterior hippocampus) repeated-measured ANOVA showed a significant four-way interaction ($F(6, 336) = 2.19, p = 0.04, \eta^2_p = 0.04$), and significant main effects of item specificity and ROI ($F(1, 56) = 9.75$ and $6.07, p = 0.003$ and $0.02, \eta^2_p = 0.15$ and 0.10), while the other main effects or interaction terms were not significant ($ps > 0.05$). To interpret the four-way interaction, we examined the results by memory type. Only for correct control, there was a significant three-way interaction among stage pair, item specificity, and ROI ($F(2, 112) = 4.21, p = 0.02, \eta^2_p = 0.07$). However, this three-way interaction was not significant for true memory, false memory, or incorrect control ($ps > 0.06$). To further interpret this three-way interaction for correct control, we examined the data by stage pair. There was a significant two-way interaction between item specificity and ROI in OP ($F(2, 56) = 4.21, p = 0.02, \eta^2_p = 0.07$), but not in OM and PM ($ps > 0.25$). Simple effect analysis showed that the neural pattern similarity for correct control in OP was marginally higher for corresponding items than non-corresponding items in the anterior hippocampus ($t(56) = 1.98, p = 0.05, d = 0.26$), but not in the posterior hippocampus ($t(56) = -0.96, p = 0.34, d = -0.13$). For true memory, although the three-way interaction among stage pair, item specificity, and ROI was not significant, there was a significant two-way interaction between item specificity and ROI ($F(1, 56) = 7.24, p = 0.009, \eta^2_p = 0.11$). Simple effect analysis showed that neural pattern similarity was higher for corresponding items than non-corresponding items in the anterior hippocampus ($t(56) = 2.52, p = 0.01, d = 0.33$), but not in the posterior hippocampus ($t(56) = -0.87, p = 0.39, d = -0.11$). For false memory and incorrect control, none of these effects or their interaction was significant, except for the main effect of item specificity for false memory ($F(1, 56) = 5.60, p = 0.02, \eta^2_p = 0.09$). These results indicated a functional dissociation along the long axis of the hippocampus for true memory and correct control rather than for false memory and incorrect control.”

2.4. There are certain sections of the paper that need more analytical detail. For example, how what the prefrontal-to-hippocampal correlation analysis conducted and how was it statistically thresholded?

Response:

We apologize for the ambiguity in the description of analytical methods in the original manuscript. We have modified them as follows.

Methods, page 42, lines 733~737:

“These contrasts were then input to a random-effects model for group analysis, with a height threshold of $Z > 2.3$ and a cluster probability of $p < 0.05$, corrected for whole-brain multiple comparisons using Gaussian Random Field Theory. Unless otherwise noted, the same threshold was used for univariate, neural pattern similarity, and correlational analyses.”

Methods, page 47, lines 837~856:

“We conducted the whole-brain voxel-wise analysis to examine whether prefrontal activity showed differential correlations with item-specific representations of original and post-event information in the hippocampus during the memory test, and whether this effect was greater for false memory than for correct control. Specifically, we examined the hippocampal-prefrontal correlation for the following contrast: (PM > OM: false memory > correct control) and (PM < OM: false memory > correct control). Because prefrontal activity might be associated with hippocampal activity rather than hippocampal representation, we used partial correlation analysis to control for the activation levels of the hippocampus during the memory test. The trial-by-trial partial correlation was calculated for false memory and correct control, separately. These coefficients were then transformed into Fisher’s Z scores for each participant, and used in the subsequent group analysis (an ordinary least square model with random effects). The threshold as described above was used.

Based the above whole-brain analysis, we then defined two prefrontal ROIs (the left and right lateral prefrontal cortex) by including all the voxels in each cluster that showed suprathreshold values for the contrast (i.e., PM > OM: false memory > correct control). The Fisher's Z scores were then extracted and averaged across voxels by condition and by ROI. We used the one-sample *t*-test to examine whether the averaged Fisher’s Z scores were significantly different from zero.”

2.5. There are well-known limitations of a searchlight analysis that should be acknowledged (see, Etzel et al., 2013, Neuroimage). I was very surprised that the hippocampal effects were not replicated in the searchlight analysis that identified the medial prefrontal effects. Some discussion of this issue is warranted.

Response:

We appreciate this reviewer pointing this out. We have added a discussion of this issue in the revised manuscript.

Discussion, page 28, lines 469~474:

“It should be noted that the hippocampal effect found in the ROI analysis was replicated in the whole-brain neural-behavioral correlation analysis, but not in the whole-brain searchlight analysis. One possible reason is that the searchlight method has several limitations, including mismatches in size and shape of relevant brain regions and those of the searchlight (Etzel et al., 2013).”

2.6. The term, neural activity pattern similarity, is very confusing and should be replaced.

Response:

We have replaced it with “neural pattern similarity” in this revised manuscript.

2.7. In the example on page 10, is the greater similarity due to the encoding of the novel information of ‘red candy box’ during the post-event phase or retrieving ‘blue candy box’ from the prior original-event phase? Why would reactivating “the man took a blue candy box” from the original event during the presentation of “the man took a red candy box” during the post-event stage lead to greater true memory? One would assume that such item-specific reactivation would lead to false memory due to the blurring/similarity of the two pieces of encoded information? More clarity is needed on the mechanisms at play here.

Response:

We agree with this reviewer that there are two possibilities, which can be tested by looking at the differences between true and false memories in cross-stage neural pattern similarity.

Possibility 1:

Cross-stage neural pattern similarity between the original-event and post-event stages is mainly due to the retrieval of original information during the post-event stage.

This possibility would be supported if cross-stage neural pattern similarity between the original-event and post-event stages was greater for true memory than false memory. For example, one participant saw the image depicting “the man took a blue candy box” during the original-event stage, and then read the misinformation “the man took a red candy box” during the post-event stage. If this participant retrieved the original information (i.e., blue candy box) while reading the misinformation, then this participant was more likely to produce a true memory than a false memory in the subsequent memory test. In this case, the cross-stage neural pattern similarity between the original-event and post-event stages was caused by the overlap between the encoding of the original information during the original-event stage (i.e., the man took a blue candy box) and the retrieval of the original information during the post-event stage (i.e., the man took a blue candy box).

Possibility 2:

Cross-stage neural pattern similarity between the original-event and post-event stages is mainly due to the blurring of the two pieces of encoded information.

This possibility would be supported if cross-stage neural pattern similarity between the original-event and post-event stages was greater for false memory than true memory. For example, one participant saw the image depicting “the man took a blue candy box” during the original-event stage, and then read the misinformation “the man took a red candy box” during the post-event stage. If this participant did not retrieve the original information (i.e., blue candy box) while reading the misinformation, then this participant was more likely to produce a false memory than a true memory in the subsequent memory test. In this case, the cross-stage neural pattern similarity between the original-event and post-event stages was caused by the overlap between the encoding of the blurring information during the original-event stage (i.e., the man took a candy box) and the encoding of the blurring information during the post-event stage (i.e., the man took a candy box).

Results of the current study support the first possibility:

Specifically, the whole-brain searchlight analysis identified multiple brain regions that showed greater item-specific neural pattern similarity between the original-event and post-event stages for true memory than false memory, including the left inferior frontal gyrus, left angular gyrus, and left middle temporal gyrus (as shown in Supplementary Figure 6 [top panel] and Supplementary Table 11 in the Supplementary Information). In contrast, no brain region showed the reversed pattern. These results suggest that cross-stage neural pattern similarity between the original-event and post-event stages is mainly due to the retrieval of original information during the post-event stage rather than the blurring of the two pieces of encoded information.

We added the above content to the captions of the Supplementary Figures 5-6 in the Supplementary Information, and added a brief discussion to clarify this mechanism in the revised manuscript. Here is the added text:

Discussion, pages 27~28, lines 450~456:

“In terms of true memory, the frontoparietal and inferior temporal cortex carried original information during the post-event and memory-test stages, as evidenced by greater OP and OM for true memory than other memory types. However, no brain region showed greater OP and OM for false memory than true memory. These results suggest that this cross-stage neural pattern similarity is mainly due to the reactivation of the original information rather than the blurring of the two pieces of encoded information.”

Caption of Figure S5 in the Supplementary Information:

“For example, one participant saw the image depicting “the man took a blue candy box” during the original-event stage (image 26), and then read the misinformation “the man took a red candy box” during the post-event stage (narrative 26). If this participant retrieved the original information (i.e., blue candy box) while reading the misinformation, then this participant was more likely to produce a true memory than a false memory in the subsequent memory test. In this case, the cross-stage neural pattern similarity between the original-event and post-event stages was caused by the overlap between the encoding of the original information during the original-event stage (i.e., the man took a blue candy box) and the retrieval of the original information during the post-event stage (i.e., the man took a blue candy box).”

Caption of Figure S6 in the Supplementary Information:

“Multiple brain regions showed greater item-specific neural pattern similarity between the original-event and post-event stages for true memory than false memory, whereas no brain region showed the reversed pattern. These results suggest that the cross-stage neural pattern similarity between the original-event and post-event stages is mainly due to the retrieval of original information during the post-event stage, rather than due to the blurring of the two pieces of encoded information.

2.8. I was surprised by the lack of an effect for the OM condition during true memory? There should be evidence of reinstatement for the correct detail (i.e., original event information) when recollecting it accurately (i.e., during the memory stage)? In contrast, it was found that OP correlated with accurate/true memory, which is surprising.

Response:

In this revision, we clarified that the lateral parietal cortex rather than the hippocampus supported true memory during the memory test. When true memory occurs, the original

information was no longer carried by the hippocampus, instead it was carried by the lateral parietal cortex during the memory test (i.e., OM). However, the hippocampus and the lateral parietal cortex have the informational connectivity in OM to produce true memory. We added the results of cortical representations and their connectivity with hippocampus in the main text.

Results, pages 21~22, lines 327~357:

“The current study focused on hippocampal representations underlying the misinformation effect. However, it is also possible that cortical representations were involved in this process. Thus, we explored whether there were other brain regions showing differences between true and false memories in item-specific representations of OM and PM. Based on the whole-brain searchlight analysis, multiple brain regions (e.g., the left angular gyrus) showed greater item-specific representations in OM for true memory than false memory, but only the posterior cingulate gyrus showed greater item-specific representations in PM for false memory than true memory (Fig. 6a). Next, we explored which brain regions carried participant-specific representations of original information to produce true memory or carried misinformation to produce false memory using the “individuation analysis”. Based on the whole-brain analysis, the lateral parietal cortex (e.g., the left angular gyrus) showed a participant-specific neural-behavioral correlation for true memory, but the medial parietal cortex (e.g., the posterior cingulate gyrus) showed a participant-specific neural-behavioral correlation for false memory (Fig. 6b and Supplementary Table 10 for details). Additional results on cortical representations and activations are shown in the Supplementary Information (Supplementary Figs. 6-11 and Supplementary Tables 11-16).

To explore the potential informational connectivity (Coutanche & Thompson-Schill, 2013; Koster et al., 2018), we assessed the covariation in trial-by-trial information (i.e., OM or PM) between the hippocampus and the cortex during the memory test. Specifically, we tested whether their trial-by-trial correlation coefficients in OM were higher for true memory than false memory, while their trial-by-trial correlation coefficients in PM were higher for false memory than true memory. Based on the whole-brain analysis with the hippocampus as a seed, the correlation coefficients between the hippocampus and several cortical regions (e.g., the left angular gyrus) in OM were higher for true memory than false memory, but the correlation coefficients between the hippocampus and the precuneus (extending to the posterior cingulate gyrus) in PM were higher for false memory than true memory (Fig. 6c). Additional brain regions revealed by this analysis are shown in Supplementary Table 17.”

Figure 6, page 23, lines 358~373:

Figure 6. Cortical representations of true and false memories and their connectivity with the hippocampus. **a** Multiple brain regions (e.g., the left angular gyrus) showed greater item-specific representations of original information during the memory test (OM) for true memory than false memory, but only the posterior cingulate gyrus showed greater item-specific representations of post-event information during the memory test (PM) for false memory than true memory. **b** Participant-specific neural-behavioral correlation. The lateral parietal cortex (e.g., the left angular gyrus) carried participant-specific original information that predicted true memory, whereas the medial parietal cortex (e.g., the posterior cingulate gyrus) carried participant-specific misinformation that predicted false memory. **c** Informational connectivity between the hippocampus and the cortex. The correlation coefficients between the hippocampus and several cortical regions (e.g., the left angular gyrus) in OM were higher for true memory than false memory, whereas the correlation coefficients between the hippocampus and the precuneus (extending to the posterior cingulate gyrus) in PM were higher for false memory than true memory in Exp. 2.

Methods, pages 48–49, lines 869–892:

“Individuation analysis for cortical representations

We conducted the whole brain voxel-wise analysis to explore whether any other brain regions carried participant-specific representations of original or post-event information that predicted behavioral performance of true memory, false memory, or correct control.

Informational connectivity between hippocampus and cortex

We conducted the whole brain voxel-wise analysis to explore whether item-specific representations in the hippocampus and those in the cortex were positively correlated. Such a correlation would indicate informational connectivity (Coutanche & Thompson-Schill, 2013; Koster et al., 2018), reflecting the functional connectivity between the hippocampus and the cortex. This analysis was based on the trial-by-trial covariation in information (item-specific representations of original or misinformation [i.e., OM or PM]) and was conducted separately for true and false memories. Spearman's correlation coefficients were used and then transferred into Fisher's Z scores. We explored which brain regions had stronger informational connectivity with the hippocampus for true memory than false memory in OM, and which regions had stronger informational connectivity with the hippocampus for false memory than true memory in PM. For this exploratory analysis, we used a relatively liberal threshold to find these regions ($Z > 1.7$ and a cluster probability of $p < 0.05$, corrected for whole-brain multiple comparison using Gaussian random field theory). Finally, the above results were masked to ensure that correlation coefficients were higher than zero for true memory in OM, and were higher than zero for false memory in PM. For example, for the contrast of true memory minus false memory in OM, results were masked with brain regions showing a positive correlation with the hippocampus for true memory in OM.”

2.9. The pattern of results for the false memory condition seems to contradict the predicted pattern of results. All three stage pairs do not differ with respect to pattern similarity (i.e., OP, PM, and OM do not statistically differ). Presumably, there should be a selective increase in the PM condition as the post-event information is being reinstated over and above the original information leading to false memory.

Response:

The pattern of results mentioned by the reviewer was based on the participant-level analysis (i.e., averaged across all trials of false memory for each participant). In this study, we conducted both participant-level and trial-level analyses because they examined two different sources of variability in the fMRI data (Chen et al., 2021). We found that the participant-level analysis did not show differences by stage pair, but the trial-level analysis did. Specifically, we found that at the trial level, when false memory occurred during the memory test, the activity in the left lateral prefrontal cortex was positively correlated with PM and negatively correlated with OM in the hippocampus. We added the new results in the revised manuscript.

Results, pages 18~19, lines 282~316:

“In support of the multiple-trace model, the above analysis suggested that there were two memory traces (i.e., OM and PM) in the hippocampus when false memory occurred during the memory test. Given the two memory traces, the prefrontal cortex is expected to be involved in selecting, suppressing, or monitoring memory traces in the hippocampus during the memory test (Fig. 5a). When false memory occurs, the neural activity in certain regions of the prefrontal cortex may be positively correlated with OM (i.e., selecting original information), or negatively correlated with PM (i.e., suppressing misinformation), or more positively correlated with PM than OM in the hippocampus (i.e., monitoring the discrepancy). Moreover, this effect should be more pronounced for false memory than for correct control, since hippocampal item-specific representations in OM and PM for correct control involve consistent information. We conducted these analyses for false memory and correct control rather than for true memory and incorrect control, because there were significant hippocampal item-specific representations in OM and PM for false memory and correct control only.

Using an exploratory whole-brain analysis and controlling for the level of hippocampal activity during the memory test, we identified two clusters located in the left lateral prefrontal cortex (MNI, -42, 28, 44, $Z = 3.71$, cluster size = 170) and the right lateral prefrontal cortex (MNI, 56, 26, 36, $Z = 3.81$, cluster size = 263) (Fig. 5b). They met the requirements that their activity showed a more positive correlation with the hippocampal item-specific representation in PM than that in OM for false memory, and that the effect described above was greater for false memory than for correct control. For the left lateral prefrontal cortex, hippocampal-prefrontal correlations for false memory were positive in PM ($t(56) = 2.41$, $p = 0.02$, $d = 0.32$) and negative in OM ($t(56) = -2.41$, $p = 0.02$, $d = -0.32$), whereas hippocampal-prefrontal correlations for correct control were non-significant in either PM or OM ($ps > 0.15$). For the right lateral prefrontal cortex, hippocampal-prefrontal correlations for false memory were positive at trend level in PM ($t(58) = 1.82$, $p = 0.07$, $d = 0.24$) and negative in OM ($t(56) = -2.75$, $p = 0.008$, $d = -0.36$), whereas hippocampal-prefrontal correlations for correct control were negative in PM ($t(56) = -2.55$, $p = 0.01$, $d = -0.34$) but not in OM ($p = 0.42$). Besides the prefrontal cortex, there was a cluster in the left superior parietal lobe (MNI, -40, -64, 26, $Z = 4.23$, cluster size = 408). No brain region showed the reversed pattern (i.e., a more positive hippocampal-prefrontal correlation in OM than that in PM for false memory compared with correct control).”

Figure 5, page 20, lines 317~326:

Figure 5. Prefrontal activity correlates with hippocampal representations when false memory occurs. **a** The hypothesis was that prefrontal activity would be more positively correlated with hippocampal representation of post-event misinformation (PM) than that of original information (OM) when false memory occurred during the memory test. **b** The lateral prefrontal activity showed a more positive correlation with hippocampal item-specific representation of post-event information than with that of original information, and this effect was more pronounced for false memory than for correct control during the memory test in Exp. 2.

2.10. Although there was a significant 3-way interaction on the pattern similarity metrics, the authors fail to conduct and report the subsidiary ANOVA's to determine which pairs of memory conditions are driving the 3-way interaction. For example, based on the follow-up tests reported, the false memory and incorrect control conditions would not yield a significant interaction and thus should be collapsed, yet the authors compare the stage pairs across each memory condition. In addition, please report all parent ANOVA results (not simply the 3-way interaction). I do not believe that the current statistical approach allows the authors to claim specificity with respect to memory condition and stage pair. For example, the authors claim that: 'The analyses reported above suggested that there were two memory traces (i.e., significant OM and PM values) in the hippocampus for false memory and correct controls during the memory test'. However, they do not directly compare the data with an ANOVA for these two memory conditions. The same applies for the following statement, 'item-specific reinstatement of post-event information (i.e., PM) in the hippocampus was greater for false memory and correct control than for true memory.' The authors need to directly compare the data from these conditions with respective ANOVA's (e.g., collapsing the false memory and correct control (assuming a null interaction), and then compared to the true memory data).

Response:

Following this reviewer's suggestion, we added all parent ANOVA results in the results section of the main text. We added a brief summary of the results of the direct comparisons in the main text, but put the details of the comparisons in the note to Supplementary Table 6 of the Supplementary Information due to limited space in the main text. The direct comparisons confirmed significant results as reported in the previous version. Here are the details.

(1) The repeated-measures ANOVA for hippocampal pattern similarity

Results, pages 13~14, lines 191~223:

"This observation was confirmed by a significant three-way interaction of the 3 (stage pairs) \times 4 (memory types) \times 2 (item specificity) repeated measures ANOVA ($F(6, 336) = 2.27, p = 0.04, \eta^2_p = 0.04$), and a significant main effect of item specificity ($F(1, 56) = 27.82, p < 0.001, \eta^2_p = 0.33$), while the other main effects and the two-way interactions were not significant in this model ($ps > 0.13$).

To probe the three-way interaction, we conducted two sets of two-way ANOVAs, by memory type and stage pair. In terms of true memory, the two-way interaction between stage pair and item specificity was significant ($F(2, 112) = 6.77, p = 0.002, \eta^2_p = 0.11$). Simple effect analysis showed that the hippocampal pattern similarity of true memory was higher for corresponding items than for non-corresponding items in OP ($t(56) = 4.33, p < 0.001, d = 0.57$) but not in OM or PM ($ps > 0.65$), and the effect of item specificity was greater in OP than in OM and PM ($t(56) = 3.06$ and $3.19, p = 0.003$ and $0.002, d = 0.41$ and 0.42). For false memory, the effect of item specificity was significant ($F(1, 56) = 16.40, p < 0.001, \eta^2_p = 0.23$), but the effect of stage pair and the interaction between stage pair and item specificity were not significant ($ps > 0.44$). These results were consistent with the multiple-trace theory (Fig. 3c). For correct control, the effect of item specificity was significant ($F(1, 56) = 17.37, p < 0.001, \eta^2_p = 0.24$), but the effect of stage pair and the interaction between stage pair and item specificity

were not significant ($ps > 0.74$). Finally, for incorrect control, none of the effects (stage pair, item specificity, or their interaction) was significant ($ps > 0.09$).

In terms of the two-way ANOVAs by stage pair, there was a significant two-way interaction between memory type and item specificity for PM ($F(3, 168) = 2.84, p = 0.04, \eta^2_p = 0.05$). Simple effect analysis showed that hippocampal pattern similarity in PM was higher for corresponding items than for non-corresponding items for false memory and correct control ($t(56) = 3.62$ and $2.83, p = 0.0006$ and $0.006, d = 0.48$ and 0.38), but not for true memory and incorrect control ($ps > 0.41$). The effect of item specificity in PM was greater for false memory and correct control than for true memory ($t(56) = 2.85$ and $2.20, p = 0.006$ and $0.03, d = 0.38$ and 0.29). In contrast, for OP or OM, the effect of item specificity was significant ($F(1, 56) = 19.98$ and $8.94, ps < 0.004, \eta^2_p = 0.25$ and 0.14), but the effect of memory type and the interaction between memory type and item specificity were not significant ($ps > 0.15$)."

Detailed results of the ANOVAs can be seen in the table below.

Table. Results of ANOVA for hippocampal pattern similarity

	df	F	p	η^2_p
3(stage pairs) × 4(memory types) × 2(item specificity)				
stage pairs	2, 112	0.35	0.70	0.006
memory types	3, 168	1.76	0.16	0.03
item specificity	1, 56	27.82	<0.001	0.33
stage pairs × memory types	6, 336	1.17	0.32	0.02
stage pairs × item specificity	2, 112	2.04	0.13	0.04
memory types × item specificity	3, 168	1.36	0.26	0.02
stage pairs × memory types × item specificity	6, 336	2.27	0.04	0.04
True memory: 3(stage pairs) × 2(item specificity)				
stage pairs	2, 112	1.45	0.24	0.03
item specificity	1, 56	7.59	0.008	0.12
stage pairs × item specificity	2, 112	6.77	0.002	0.11
False memory: 3(stage pairs) × 2(item specificity)				
stage pairs	2, 112	0.83	0.44	0.01
item specificity	1, 56	16.40	<0.001	0.23
stage pairs × item specificity	2, 112	0.76	0.47	0.01
Correct control: 3(stage pairs) × 2(item specificity)				
stage pairs	2, 112	0.20	0.82	0.004
item specificity	1, 56	17.37	<0.001	0.24
stage pairs × item specificity	2, 112	0.31	0.74	0.005
Incorrect control: 3(stage pairs) × 2(item specificity)				
stage pairs	2, 112	0.35	0.70	0.006
item specificity	1, 56	3.05	0.09	0.05
stage pairs × item specificity	2, 112	1.23	0.30	0.02
OP: 4(memory types: TM, FM, CC, IC) × 2(item specificity)				
memory types	3, 168	0.18	0.91	0.003
item specificity	1, 56	19.98	<0.001	0.25
memory types × item specificity	3, 168	1.30	0.28	0.02
OM: 4(memory types: TM, FM, CC, IC) × 2(item specificity)				
memory types	3, 168	1.21	0.31	0.02
item specificity	1, 56	8.94	0.004	0.14
memory types × item specificity	3, 168	1.80	0.15	0.03
PM: 4(memory types: TM, FM, CC, IC) × 2(item specificity)				
memory types	3, 168	2.63	0.052	0.04
item specificity	1, 56	8.81	0.004	0.14
memory types × item specificity	3, 168	2.84	0.04	0.05

Note. Bolded scores represent significant *p* values. OP: Neural pattern similarity between original-event and post-event stages. OM: Neural pattern similarity between original-event and memory-test stages. PM: Neural pattern similarity between post-event and memory-test stages. TM: true memory of critical items. FM: false memory of critical items. CC: correct control items. IC: incorrect control items.

(2) Additional analyses of hippocampal pattern similarity comparing true memory with false memory and correct control

Based on this reviewer's suggestion, we made a direct comparison between false memory and correct control, and then combined them for a comparison with true memory. In this revision, we added it in the main text and the note of Supplementary Table 6 of the Supplementary Information.

Results, pages 14~15, lines 223~232:

“In addition, we directly compared false memory with correct control by conducting a 2 (memory types: false memory vs. correct control) × 3 (stage pairs) × 2 (item specificity) repeated measures ANOVA. Since none of the interaction terms was significant, we combined false memory and correct control and compared them with true memory using a 2 (memory types: [false memory & correct control] vs. true memory) × 3 (stage pairs) × 2 (item specificity) repeated measures ANOVA. These analyses confirmed that the effect of item specificity in PM in the hippocampus was greater for false memory and correct control than for true memory (Details are shown in Supplementary Table 6).”

Supplementary Information, note of Supplementary Table 6:

“Additional results of hippocampal pattern similarity:

To directly compare false memory and correct control, we conducted a 2 (memory types: false memory vs. correct control) × 3 (stage pairs) × 2 (item specificity) repeated measures ANOVA. Results showed that only the main effect of item specificity was significant ($F(1, 56) = 6.79, p < 0.001, \eta^2_p = 0.32$), while none of the other main effects or interaction terms was significant ($ps > 0.39$). Thus, we combined false memory and correct control, and compared them with true memory using the 2 (memory types: [false memory & correct control] vs. true memory) × 3 (stage pairs) × 2 (item specificity) repeated measures ANOVA. Results showed a significant three-way interaction ($F(2, 112) = 6.31, p = 0.003, \eta^2_p = 0.10$), a significant two-way interaction between stage pair and item specificity ($F(2, 112) = 3.39, p = 0.04, \eta^2_p = 0.06$), and significant main effects of item specificity and memory type ($F(1, 56) = 25.94$ and $6.15, ps < 0.02, \eta^2_p = 0.32$ and 0.10), while the main effect of stage pair and the other two-way interactions were not significant in this model ($ps > 0.06$). To probe this three-way interaction, we conducted two-way ANOVA by stage pair. Only for PM, there were a significant two-way interaction between memory type and item specificity ($F(1, 56) = 7.79, p = 0.007, \eta^2_p = 0.12$), and significant main effects of memory type and item specificity ($F(1, 56) = 6.76$ and $5.10, p = 0.01$ and $0.03, \eta^2_p = 0.11$ and 0.08). Simple effect analysis for PM showed that hippocampal pattern similarity was higher for corresponding items than for non-corresponding items in the combined memory type of false memory and correct control ($t(56) = 4.29, p < 0.001, d = 0.57$), but not in true memory ($p = 0.82$). For OP, the main effect of item specificity was significant ($F(1, 56) = 23.82, p < 0.001, \eta^2_p = 0.30$), but the main effect of memory type and the interaction were not significant ($ps > 0.05$). For OM, there were significant main effects of item specificity ($F(1, 56) = 4.48, p = 0.04, \eta^2_p = 0.07$ [corresponding > non-corresponding]) and memory type ($F(1, 56) = 7.41, p = 0.009, \eta^2_p = 0.12$ [false memory & correct control > true memory]), but their interaction was not significant ($p = 0.09$).”

2.11. It is necessary to provide the pattern similarity data separately for the corresponding and non-corresponding trials. The authors only provide the difference data. It is unknown whether the greater than zero item-specific reinstatement is due to a decrease from zero for the non-corresponding trials or an increase for corresponding trials. Obviously, it should be the former to support the logic of the analysis.

Response:

In the original manuscript, we reported the means and standard deviations of hippocampal pattern similarity across the three stages of each memory type for corresponding items, non-corresponding items, and their differences (i.e., item-specific representations) in Supplementary Table 6 of the Supplementary Information. In this revision, based on this reviewer's suggestion, we showed data for corresponding and non-corresponding items in Figure 3d in the main text. It should be noted that, as requested by the editor, the original bar graph was replaced by the split violin graph. It shows that the current findings were mainly driven by data from corresponding items rather than from non-corresponding items, which supports the logic of the analysis.

Results, page 12, Figure 3d:

Figure 3d. Hippocampal pattern similarity for the corresponding items and for the non-corresponding items for three different stage pairs by memory type (true memory, false memory, correct control, and incorrect control) in Exp. 2. Data are visualized as split-violin plots with bounds indicating the min and max values and the black dots showing the means. Error bars indicate within-participant SEs. **, $p < 0.01$.

2.12. The individuation analysis is quite interesting. One would like to see a similar analysis done to predict true memory. Such a finding would support models of hippocampal function during episodic memory (i.e., reinstatement of item-specific information supports veridical recollection; Chadwick et al., 2010, Neuron; Tompary et al., 2016, Hippocampus; Norman & O'Reilly, 2003; Xue, 2018, TICS).

Response:

We thank Reviewer 2 for appreciating the individuation analysis and providing valuable references. In this revision, we added the analysis of predicting true memory in the methods and results sections and Supplementary Tables 9-10 in the Supplementary Information, and we also added these references in the discussion section.

(1) Individuation analysis of hippocampal representations for true memory

Based on this reviewer's suggestion, we added the hippocampal results of the individuation analysis for true memory in the results section and Supplementary Table 9 in the Supplementary Information and added a brief discussion in this revision.

Results, page 16, lines 261~266:

“In the hippocampus, there was no participant-specific mapping between OM and true memory ($t(56) = -1.55$, $p = 0.13$, $d = -0.21$), but the magnitude of negative correlations between PM and true memory was significantly smaller in within-participant than between-participant analyses ($t(56) = -2.32$, $p = 0.02$, $d = -0.31$, Supplementary Table 9 for details).”

Supplementary Table 9. The means and standard deviations (Mean \pm SD) for the Fisher's Z scores of within-participant and between-participant correlations between hippocampal item-specific representation and behavioral performance of false memory, correct control, or true memory in Exp. 2.

	Correlations with PM		Correlations with OM	
	Within -participant	Between -participant	Within -participant	Between -participant
False memory	0.022 \pm 0.067	-0.005 \pm 0.056	0.006 \pm 0.063	-0.002 \pm 0.038
Correct control	0.011 \pm 0.084	0.005 \pm 0.056	0.013 \pm 0.073	0.004 \pm 0.050
True memory	-0.019 \pm 0.066	-0.002 \pm 0.046	-0.009 \pm 0.067	0.003 \pm 0.039

Note. OM: Neural pattern similarity between original-event and memory-test stages.

PM: Neural pattern similarity between post-event and memory-test stages.

Item-specific neural pattern similarity was calculated by the neural pattern similarity for the corresponding items minus that for the non-corresponding items in the same event for each memory type. Greater within-participant than between-participant neural-behavioral correlations are shown in bold. Lower within-participant than between-participant neural-behavioral correlations are shown in italics. As shown by a previous study (Chadwick et al., 2016), the results of individuation analysis are meaningful only when the within-participant correlation is greater than the between-participant correlation. The negative within-participant correlation for true memory contradicts the premise of the individuation analysis. Therefore, we are not concerned with this negative correlation.

Discussion, page 28, lines 459~461:

“Unlike the typical role of the hippocampus in predicting true memory (Chadwick et al., 2010; Norman & O'Reilly, 2003; Tompary et al., 2016; Xue, 2018), the post-event misinformation altered the role of the hippocampus without changing the role of the lateral parietal cortex in true memory.”

(2) Individuation analysis of cortical representations predicting memories

In addition, we added the cortical results of the individuation analysis in the results section and Supplementary Table 10 in the Supplementary Information and added a brief discussion in this revision. Details can be seen in the revised manuscript and in our response to Comment 2.8 above.

2.13. As a whole greater clarity is needed across the different conditions and what comparisons support what predictions/patterns across the three theoretical models.

Response:

In this revision, we clarified that the three theoretical models (i.e., no-retention, trace-alternation, and multiple-trace), predictions/hypotheses derived from them, and comparisons needed to test them (Figures 1 and 3c in the manuscript).

Introduction, page 4, lines 46~48:

“Three theoretical perspectives have been proposed to explain false memory induced by misinformation: non-retention, trace-alteration, and multiple-trace models (see Fig. 1 for a schematic representation of these models).”

Figure 3c shows the hypothesized outcomes of false memory based on the three theoretical models. The non-retention model posits item-specific representations (i.e., corresponding > non-corresponding) for PM but not for OP and OM, the trace-alteration model posits item-specific representations for OP and PM but not for OM, and the multiple-trace model posits item-specific representations in the hippocampus for all three stage pairs of false memory.

Results, page 12, lines 181~182, Figure 3c:

Figure 3. c Hypothesized outcomes of false memory based on the three theories.

Results, pages 13~14, lines 204~210:

“For false memory, the effect of item specificity was significant ($F(1, 56) = 16.40, p < 0.001, \eta^2_p = 0.23$), but the effect of stage pair and the interaction between stage pair and item specificity were not significant ($ps > 0.44$). These results were consistent with the multiple-trace theory (Fig. 3c). For correct control, the effect of item specificity was significant ($F(1, 56) = 17.37, p < 0.001, \eta^2_p = 0.24$), but the effect of stage pair and the interaction between stage pair and item specificity were not significant ($ps > 0.74$).”

Moreover, the related discussion is added to the discussion section (pages 25~26, lines 397~428).

2.14. Please add bars denoting which stage pairs (OP, OM, PM) statistically differ for each memory condition (true, false, correct control, and incorrect control).

Response:

Following this suggestion, we added bars and asterisks to show the significance of the comparisons of interest, using ** indicating $p < 0.01$, but due to the space limitation of this figure, we reported other significant results in the main text (Results, page 13, lines 203~204).

Reply to Reviewer 3's comments:

The manuscript by Shao and colleagues describes a pair of human experiments (behavioral and fMRI) investigating the misinformation effect. The experiments use a standard experimental structure where events are first encoded (original memories), then subjects are subsequently exposed (in a post-event stage) to information that is consistent, neutral, or inconsistent (misinformation) with the original memory and finally memory for the original event is tested. As is standard, they report a behavioral effect where subjects are more likely to endorse false information when they have been exposed to that (mis)information during the post-encoding phase. While the behavioral effects are simply a replication of a well-established finding, the experimental materials used here are quite nice in that the events were designed to have a lot of individual elements which could then be cued and tested—something that is important for conducting an adequately powered fMRI study. For the fMRI experiment, the main question is whether the tendency to mistakenly 'remember' the misinformation (i.e., false memories) is related to hippocampal pattern similarity across the three memory stages: the original event, the post-event stage, and the memory test. The key and most interesting finding is that true vs. false memories are associated with different patterns of hippocampal pattern similarity across stages. In relative terms, similarity between the original event and the post-event stage is greater for true than false memories whereas similarity between the post-event stage and the final memory test is greater for false than true memories. This pattern of results is consistent with an interpretation wherein reinstatement of the original event during the post-event stage helps prevent falsely remembering the misinformation whereas reinstatement of the misinformation during the final memory test has the opposite effect (i.e., it leads to false memories). A second analysis shows that there is a subject-specific 'alignment' of the fMRI pattern similarity results with the specific elements that are behaviorally mis-remembered. Finally, it is shown that prefrontal activity is negatively correlated with the reinstatement of the original memory during the final test, which is interpreted as reflecting inhibition of the original memory.

There are a number of appealing elements to this paper and it is generally well written and clearly presented. In particular, it is a nice experimental design with a robust sample size and with a clear presentation of competing theoretical accounts and how the current experiment can test these. The first two findings are very nice and I think they will be of relatively broad interest. The last finding (related to 'inhibition' of true memories) was less compelling to me. The main weaknesses in the paper are (1) while I do think the current paper represents an advance, there are, as the authors note, several prior papers which have addressed conceptually similar ideas—though, with less sophisticated approaches and (2) I think the paper gets into a bit of trouble by tending to present interpretations/assumptions instead of results and by failing to consider alternative accounts. Taken together, while my overall impression is somewhat favorable, there are some notable weaknesses in the current form of the manuscript.

Response:

We thank Reviewer 3 for his/her very encouraging comments and insightful suggestions. In this revision, we clarified the novelty and importance of this study in the discussion (pages 30~31, lines 518~521). Moreover, we corrected the over-interpretation and made the following revisions accordingly.

Comments.

3.1. I believe the biggest issue in the manuscript is over-interpretation of the findings. There are three main examples of this.

a. First, fMRI pattern similarity is taken as evidence of reactivation without any consideration of alternative explanations. However, item-specific similarity could also easily be explained by the fact that within-item comparisons tend to have a lot of shared semantic information. So, even if there was no memory component whatsoever, I would expect some degree of similarity (greater for within than across items) just because of shared semantic information. To be fair, I would not really expect a “semantic only” effect in the hippocampus, but the point is that the similarity measure is not a pure test of episodic memory because it is confounded with semantic similarity. As such, rather than describing the results as “reinstatement,” it would be more appropriate to describe the results more neutrally as “pattern similarity” (since this is what is actually measured). Obviously, higher pattern similarity might reflect reinstatement, but that is an interpretation. To be clear, I do think that the most likely explanation of the pattern similarity effects is a “reinstatement account”—but I just think that there should be a distinction between the result and the interpretation.

Response:

In the revision, we used “pattern similarity” instead of “reinstatement” in the results section.

b. Similarly, it is argued (e.g., Line 231) that the pattern similarity analyses suggest “that there were two memory traces (i.e., significant OM and PM values).” And elsewhere in the paper it is more or less taken as fact that there were “competing” memory traces. But, again, this is an interpretation. From my perspective, I did not find the evidence of competing memory traces to be compelling. In particular, the PM value could, in principle, reflect reinstatement of the exact same information that is reflected in the OP measure. In fact, one way of looking at the data is that PM similarity does not exceed OM similarity for any of the conditions. In other words, the representation at the memory test is no more similar to the post-event stage than it is to the original event. Obviously, the fact that PM similarity is greater for false memories fits nicely with the interpretation that there is a competing trace (of the misinformation), but, again, that is an interpretation. An alternative interpretation, for example, could be that if the ‘wrong’ information from the original event is reinstated at the post-event stage and then again at the final memory test (e.g., remembering a feature of the original event that is not related to the misinformation), it results in a false memory. I am not saying I find this account more likely, but the key point is that I think there is a plausible account where the relatively high OM/PM similarity in the false memory condition reflects reinstatement of SOMETHING from the original memory as opposed to reinstatement of the misinformation. To make a stronger claim about distinct memory traces, I think it would be critical to statistically test whether the post-event representation explains unique variance in the memory-test representation (that is, variance that is not explained by the original event representation).

Response:

We thank Reviewer 3 for raising this important issue. In the results section, we deleted the term "competing memory traces" and revised it to "two memory traces".

Results, page 18, lines 282~284:

"In support of the multiple-trace model, the above analysis suggested that there were two memory traces (i.e., OM and PM) in the hippocampus when false memory occurred during the memory test."

Following the reviewer's suggestion, we used partial correlation analyses to test whether the post-event representation explained unique variance in the memory-test representation (i.e., variance that was not explained by the original-event representation). We added these results in the revision.

Results, page 15, lines 232~235:

"Furthermore, the results described above were not affected by the univariate activation level in the hippocampus, and were unchanged when using partial correlations for PM and OM and after correction for correlation comparisons (Details are shown in Supplementary Tables 6-7)."

Note of Supplementary Table 6 in the Supplementary Information:

"Additional results using partial correlations for PM and OM:

Partial correlations were calculated for PM and OM at the trial level for corresponding and non-corresponding items, separately. For the corresponding items, a partial correlation for PM was calculated between the neural pattern of one item during the memory-test stage (e.g., test item 26) and the neural pattern of its corresponding narrative during the post-event stage (e.g., narrative 26), after controlling for the neural pattern of its corresponding image during the original-test stage (e.g., image 26). For the non-corresponding items in the same event, partial correlations for PM were calculated between neural patterns during the memory-test (e.g., test item 26) and post-event stages (e.g., narrative 1), after controlling for that during the original-test stage

(e.g., image 1). Then, these similarity scores were transformed into Fisher's Z scores, which were then averaged to generate the neural pattern similarity value for each type of trial. The same method was used to calculate partial correlations for OM, except that neural pattern similarities were computed between original-event and memory-test stages after controlling for the neural pattern during the post-event stage. It should be noted that we did not calculate partial correlations for OP, because the memory test was conducted after the original and post-event stages (i.e., OP was unlikely to be influenced by neural patterns during the subsequent memory test). Using the partial correlations for PM and OM and original correlations for OP, we conducted the 3 by 4 by 2 repeated measures ANOVA on the hippocampal pattern similarity.

These additional analyses confirmed the significant results as reported in Figure 3d in the main text. A 3 (stage pairs) \times 4 (memory types) \times 2 (item specificity) repeated measures ANOVA showed a significant three-way interaction ($F(6, 336) = 2.15, p = 0.047, \eta^2_p = 0.04$), and a significant main effect of item specificity ($F(1, 56) = 25.70, p < 0.001, \eta^2_p = 0.31$), while the other main effects and the two-way interactions were not significant ($ps > 0.13$). We then analyzed the data by memory type. For true memory, the two-way interaction between stage pair and item specificity was significant ($F(2, 112) = 5.86, p = 0.004, \eta^2_p = 0.09$). Simple effect analysis for true memory showed that the hippocampal pattern similarity was higher for corresponding items than for non-corresponding items in OP ($t(56) = 4.33, p < 0.001, d = 0.57$) but not in OM or PM ($ps > 0.77$), and the effect of item specificity was greater in OP than in OM and PM ($t(56) = 3.22$ and $2.84, p = 0.002$ and $0.006, d = 0.43$ and 0.38). For false memory, the effect of item specificity was significant ($F(1, 56) = 17.61, p < 0.001, \eta^2_p = 0.24$), but the effect of stage pair and the interaction between stage pair and item specificity were not significant ($ps > 0.46$). For correct control, the effect of item specificity was significant ($F(1, 56) = 14.60, p < 0.001, \eta^2_p = 0.21$), but the effect of stage pair and the interaction between stage pair and item specificity were not significant ($ps > 0.76$). Finally, for incorrect control, none of the effects (stage pair, item specificity, or their interaction) was significant ($ps > 0.06$). For PM, hippocampal pattern similarity was higher for corresponding items than that for non-corresponding items for false memory ($t(56) =$

3.48, $p = 0.0009$, $d = 0.46$) but not for true memory ($p = 0.97$), and the effect of item specificity in PM was greater for false memory than for true memory ($t(56) = 2.60$, $p = 0.01$, $d = 0.34$). These additional results still support the multiple-trace theory (i.e., two distinct memory traces during the memory test).”

We also present the original results (top panel) and additional results (bottom panel) in the following figure.

c. Finally, the idea that prefrontal cortex inhibited the original memory trace feels like a stretch to me. Frankly, I really do not know what to make of the prefrontal data. The argument about inhibition feels a bit backwards to me. Namely, it seems more intuitive that the prefrontal cortex would play a role in inhibiting the (incorrect) post-event memory as opposed to the original memory. Thus, I found the prefrontal correlation results to be somewhat puzzling and did not find the inhibition interpretation intuitive or compelling.

Response:

We apologize for the confusion. As suggested by Reviewers 2 and 3 (i.e., comments 2.1, 2.9, 3.1c, and 3.2), we reanalyzed the hippocampal-prefrontal correlations. Briefly, we found that the lateral prefrontal activity showed a more positive correlation with hippocampal item-specific representation of post-event information than with that of original information, and this effect was more pronounced for false memory than for correct control during the memory test. The revised results are included in the manuscript and in our response to Reviewer 2's Comment 2.9. Here is the revised discussion.

Discussion, pages 26~27, lines 428~440:

“When false memory was produced during the memory test, the hippocampus carried not only the corresponding post-event misinformation, but also the corresponding original information. Since there were two memory traces for false memory during retrieval, this discrepancy would trigger the source monitoring process in the lateral prefrontal cortex (Dobbins et al., 2003; Mitchell & Johnson, 2009; Simons & Spiers, 2003; Ye et al., 2016; Zhu et al., 2019). Indeed, our results suggest that the strong memory traces of misinformation but weak memory traces of original information in the hippocampus triggered monitoring processes for false memory in the lateral prefrontal cortex (i.e., positively correlated with PM but negatively correlated with OM). However, we did not observe any positive hippocampal-prefrontal correlation for OM (i.e., selecting original information), or any negative correlation for PM (i.e., suppressing misinformation). Overall, our findings indicate that the lateral prefrontal cortex works in concert with the hippocampus to resolve this conflict during the memory test when false memory occurs.”

3.2. For the prefrontal correlation analyses, it is reported that similar effects were not observed for the control condition, but a stronger test would be to actually test whether effects were statistically greater for the false memory than control condition. But this would have to be done within the searchlight analysis.

Response:

We tested whether these effects were statistically greater for the false memory than correct control in the whole-brain analysis. In short, results showed that the lateral prefrontal activity showed a statistically more positive correlation with hippocampal item-specific representation of post-event information than with that of original information, and this effect was statistically more pronounced for false memory than for correct control during the memory test. Details can be seen in the main text (pages 18~20, 47) and in our response to Reviewer 2's Comment 2.9.

3.3. For the second main question (starting on Line 230), the question is framed as whether “the results reflected reinstatements of post-event information shared among all research participants, or there might be consistent individual differences.” This framing was quite confusing to me. Ultimately, I understand the analysis and I liked the result, but I think this could be framed in a more effective way.

Response:

We apologize for this ambiguity. We revised the sentence as follows.

Results, page 15, lines 245~247:

“There would be a participant-specific mapping between the hippocampal item-specific representation of post-event information and the behavioral pattern of false memory.”

3.4. In initially reading the paper, I was a bit surprised by the lack of consideration of any region other than the hippocampus. This is only addressed at the very end of the Results where the reader is oriented to the supplement. I do appreciate that the supplement reports whole-brain analyses. However, it's a bit hard to get a full sense of how the effects in the hippocampus compare to those in other regions. From what I can tell, it does seem that the pattern of results in many cortical areas is qualitatively dissimilar to the hippocampus. While that is not problematic in and of itself, it does complicate some of the conclusions. That is, many of the conclusions in the paper are framed as "general" conclusions about the basis of the misinformation effect; but, really, they are specific to the hippocampus. For example, it appears from Figure S7 that quite a few regions show stronger OM similarity for true vs. false memories. In fact, OM similarity looks like it may be a more robust predictor of true memories (in terms of number of regions) than the other similarity measures. This contrasts with the data from the hippocampus and, in fact, may suggest a mechanistically distinct way in which false memories are avoided. While I appreciate that the authors have an a priori interest in the hippocampus—and I think that makes sense—I do think that some conclusions in the paper need to be more carefully presented given that the pattern of data from the hippocampus appears to qualitatively differ from many other brain regions.

Response:

Following the reviewers' suggestions, we put the results of cortical representations and the results of informational connectivity with the hippocampus in the main text (pages 21~23). Also see our response to Reviewer 2's Comments 2.8. We added the following content in the discussion section.

Discussion, pages 27~29, lines 449~485:

“Outside the hippocampus, an exploratory searchlight analysis showed that cortical representations also contributed to true and false memories. In terms of true memory, the frontoparietal and inferior temporal cortex carried original information during the post-event and memory-test stages, as evidenced by greater OP and OM for true memory than other memory types. However, no brain region showed greater OP and OM for false memory than true memory. These results suggest that this cross-stage neural pattern similarity is mainly due to the reactivation of the original information rather than the blurring of the two pieces of encoded information. Among these brain regions, the lateral parietal cortex carried participant-specific neural representations of original information that predicted true memory during the memory test. Unlike the typical role of the hippocampus in predicting true memory (Chadwick et al., 2010;

Norman & O'Reilly, 2003; Tompary et al., 2016; Xue, 2018), the post-event misinformation altered the role of the hippocampus without changing the role of the lateral parietal cortex in true memory. Although the lateral parietal cortex rather than the hippocampus carried original information during the memory test, the original information was communicated between these two brain regions, as evidenced by the stronger informational connectivity between the hippocampus and the left angular gyrus in OM for true memory than false memory.

False memory was predicted by the participant-specific representations of misinformation carried by the hippocampus and medial parietal cortex (e.g., posterior cingulate gyrus), which communicated with each other during the memory test. It should be noted that the hippocampal effect found in the ROI analysis was replicated in the whole-brain neural-behavioral correlation analysis, but not in the whole-brain searchlight analysis. One possible reason is that the searchlight method has several limitations, including mismatches in size and shape of relevant brain regions and those of the searchlight (Etzel et al., 2013). These findings were consistent with previous studies showing that lateral and medial parietal cortices differ in their role in memory representation during retrieval (Ritchey & Cooper, 2020; Staresina & Wimber, 2019). In the misinformation paradigm, true memory is associated with the neural representation in the left angular gyrus as it retrieves original-event episodic details (Bonnici et al., 2016; Kuhl & Chun, 2014; Rugg & King, 2018; Thakral et al., 2017); whereas false memory is associated with the neural representation in the posterior cingulate gyrus as it links post-event narratives with prior knowledge (Binder et al., 2009; Bird et al., 2015; Gurguryan & Sheldon, 2019). Extending these previous studies, our study showed that each individual's unique neural representation of original or post-event information in these brain regions predicted idiosyncratic patterns of true or false memory. In line with the role of the posterior medial network (Ritchey & Cooper, 2020), our findings suggest that the lateral and medial parietal cortices maintain distinct hippocampal-cortical communications of original-event and post-event information to produce true and false memories.”

Discussion, pages 30~31, lines 513~521:

“In conclusion, our research provides direct evidence that dynamic changes in human hippocampal representations across three memory stages underlie false memory from misinformation. These neuroimaging findings support the multiple-trace memory theory and source monitoring framework of misinformation false memory. Moreover, the hippocampus works with the lateral and medial parietal cortices to produce true and false memories, respectively. These findings enhance our understanding of the neural mechanisms underlying the reconstructive nature of human memory by providing an integrated model of the hippocampal-cortical network involved in false memory.”

References

- Badre, D., & Wagner, A. D. (2004). Selection, integration, and conflict monitoring: Assessing the nature and generality of prefrontal cognitive control mechanisms. *Neuron, 41*(3), 473-487.
- Badre, D., & Wagner, A. D. (2007). Left ventrolateral prefrontal cortex and the cognitive control of memory. *Neuropsychologia, 45*(13), 2883-2901.
- Binder, J. R., Desai, R. H., Graves, W. W., & Conant, L. L. (2009). Where is the semantic system? A critical review and meta-analysis of 120 functional neuroimaging studies. *Cerebral Cortex, 19*(12), 2767-2796.
- Bird, C. M., Keidel, J. L., Ing, L. P., Horner, A. J., & Burgess, N. (2015). Consolidation of complex events via reinstatement in posterior cingulate cortex. *Journal of Neuroscience, 35*(43), 14426-14434.
- Bonnici, H. M., Richter, F. R., Yazar, Y., & Simons, J. S. (2016). Multimodal feature integration in the angular gyrus during episodic and semantic retrieval. *Journal of Neuroscience, 36*(20), 5462-5471.
- Carpenter, A. C., Thakral, P. P., Preston, A. R., & Schacter, D. L. (2021). Reinstatement of item-specific contextual details during retrieval supports recombination-related false memories. *NeuroImage, 236*, 118033.
- Chadwick, M. J., Anjum, R. S., Kumaran, D., Schacter, D. L., Spiers, H. J., & Hassabis, D. (2016). Semantic representations in the temporal pole predict false memories. *Proceedings of the National Academy of Sciences, 113*(36), 10180-10185.

- Chadwick, M. J., Hassabis, D., Weiskopf, N., & Maguire, E. A. (2010). Decoding individual episodic memory traces in the human hippocampus. *Current Biology*, *20*(6), 544-547.
- Cohn-Sheehy, B. I., Delarazan, A. I., Reagh, Z. M., Crivelli-Decker, J. E., Kim, K., Barnett, A. J., Zacks, J. M., & Ranganath, C. (2021). The hippocampus constructs narrative memories across distant events. *Current Biology*, *31*(22), 4935-4945.e4937.
- Coutanche, M., & Thompson-Schill, S. (2013). Informational connectivity: Identifying synchronized discriminability of multi-voxel patterns across the brain. *Frontiers in Human Neuroscience*, *7*.
- Dobbins, I. G., Rice, H. J., Wagner, A. D., & Schacter, D. L. (2003). Memory orientation and success: Separable neurocognitive components underlying episodic recognition. *Neuropsychologia*, *41*(3), 318-333.
- Eichenbaum, H. (2017). Memory: Organization and control. *Annual Review of Psychology*, *68*(1), 19-45.
- Etzel, J. A., Zacks, J. M., & Braver, T. S. (2013). Searchlight analysis: Promise, pitfalls, and potential. *NeuroImage*, *78*, 261-269.
- Gurguryan, L., & Sheldon, S. (2019). Retrieval orientation alters neural activity during autobiographical memory recollection. *NeuroImage*, *199*, 534-544.
- Henriksson, L., Khaligh-Razavi, S.-M., Kay, K., & Kriegeskorte, N. (2015). Visual representations are dominated by intrinsic fluctuations correlated between areas. *NeuroImage*, *114*, 275-286.

- Karanian, J. M., Rabb, N., Wulff, A. N., Torrance, M. G., Thomas, A. K., & Race, E. (2020). Protecting memory from misinformation: Warnings modulate cortical reinstatement during memory retrieval. *Proceedings of the National Academy of Sciences*, *117*(37), 22771-22779.
- Koster, R., Chadwick, M. J., Chen, Y., Berron, D., Banino, A., Düzel, E., Hassabis, D., & Kumaran, D. (2018). Big-loop recurrence within the hippocampal system supports integration of information across episodes. *Neuron*, *99*(6), 1342-1354.e1346.
- Kuhl, B. A., & Chun, M. M. (2014). Successful remembering elicits event-specific activity patterns in lateral parietal cortex. *Journal of Neuroscience*, *34*(23), 8051-8060.
- Loftus, E. F. (2005). Planting misinformation in the human mind: A 30-year investigation of the malleability of memory. *Learning & Memory*, *12*(4), 361-366.
- Loftus, E. F., Miller, D. G., & Burns, H. J. (1978). Semantic integration of verbal information into a visual memory. *Journal of Experimental Psychology: Human Learning and Memory*, *4*(1), 19-31.
- Maguire, E. A. (2022). Does memory research have a realistic future? *Trends in Cognitive Sciences*, *26*(12), 1043-1046.
- Mitchell, K. J., & Johnson, M. K. (2009). Source monitoring 15 years later: What have we learned from fMRI about the neural mechanisms of source memory? *Psychological Bulletin*, *135*(4), 638-677.
- Mitchell, K. J., Johnson, M. K., Raye, C. L., & Greene, E. J. (2004). Prefrontal cortex activity associated with source monitoring in a working memory task. *Journal of Cognitive Neuroscience*, *16*(6), 921-934.

- Norman, K. A., & O'Reilly, R. C. (2003). Modeling hippocampal and neocortical contributions to recognition memory: A complementary-learning-systems approach. *Psychological Review*, 110(4), 611-646.
- Okado, Y., & Stark, C. E. (2005). Neural activity during encoding predicts false memories created by misinformation. *Learning & Memory*, 12(1), 3-11.
- Poppenk, J., Evensmoen, H. R., Moscovitch, M., & Nadel, L. (2013). Long-axis specialization of the human hippocampus. *Trends in Cognitive Sciences*, 17(5), 230-240.
- Preston, A. R., & Eichenbaum, H. (2013). Interplay of hippocampus and prefrontal cortex in memory. *Current Biology*, 23(17), 764-773.
- Putnam, A. L., Sungkhasettee, V. W., & Roediger, H. L. (2017). When misinformation improves memory: The effects of recollecting change. *Psychological Science*, 28(1), 36-46.
- Ritchey, M., & Cooper, R. A. (2020). Deconstructing the posterior medial episodic network. *Trends in Cognitive Sciences*, 24(6), 451-465.
- Rugg, M. D., & King, D. R. (2018). Ventral lateral parietal cortex and episodic memory retrieval. *Cortex*, 107, 238-250.
- Simons, J. S., & Spiers, H. J. (2003). Prefrontal and medial temporal lobe interactions in long-term memory. *Nature Reviews Neuroscience*, 4(8), 637-648.
- Staresina, B. P., & Wimber, M. (2019). A neural chronometry of memory recall. *Trends in Cognitive Sciences*, 23(12), 1071-1085.

- Stark, C. E., Okado, Y., & Loftus, E. F. (2010). Imaging the reconstruction of true and false memories using sensory reactivation and the misinformation paradigms. *Learning & Memory, 17*(10), 485-488.
- Thakral, P. P., Madore, K. P., & Schacter, D. L. (2017). A role for the left angular gyrus in episodic simulation and memory. *Journal of Neuroscience, 37*(34), 8142-8149.
- Tomparry, A., Duncan, K., & Davachi, L. (2016). High-resolution investigation of memory-specific reinstatement in the hippocampus and perirhinal cortex. *Hippocampus, 26*(8), 995-1007.
- Vilberg, K. L., & Rugg, M. D. (2012). The neural correlates of recollection: Transient versus sustained fMRI effects. *Journal of Neuroscience, 32*(45), 15679-15687.
- Xue, G. (2018). The neural representations underlying human episodic memory. *Trends in Cognitive Sciences, 22*(6), 544-561.
- Ye, Z., Zhu, B., Zhuang, L., Lu, Z., Chen, C., & Xue, G. (2016). Neural global pattern similarity underlies true and false memories. *Journal of Neuroscience, 36*(25), 6792-6802.
- Zhu, B., Chen, C., Loftus, E. F., He, Q., Lei, X., Dong, Q., & Lin, C. (2016). Hippocampal size is related to short-term true and false memory, and right fusiform size is related to long-term true and false memory. *Brain Structure and Function, 221*(8), 4045-4057.
- Zhu, B., Chen, C., Shao, X., Liu, W., Ye, Z., Zhuang, L., Zheng, L., Loftus, E. F., & Xue, G. (2019). Multiple interactive memory representations underlie the induction of false memory. *Proceedings of the National Academy of Sciences, 116*(9), 3466-3475.

Reviewer #1 (Remarks to the Author):

Issue 1.1

Thank you for addressing my major concern and adding this as a limitation of the current study. I agree that the difference between true and false memory in the neural pattern similarity observed in the hippocampus (everything else being equal) is "likely" to be explained by a memory representation being reinstated rather than re-processing of scenes elicited by words. However, this conclusion cannot be taken for granted given that it is always possible that by chance we observe greater or smaller neural pattern similarity for different conditions. For example, how do we interpret the greater than zero neural pattern similarity in the correct control condition within the hippocampus? Is that due to re-processing of the original scene or is it a memory effect?

Issue 1.2

Thank you for performing an additional analysis. I would like to point out that the existence of the original and post-event memory traces in two different brain regions (lateral prefrontal cortex and hippocampus, respectively) is only exploratory since the former was only found after performing a searchlight analysis that was carried out subsequently. Thus, the authors should highlight in the discussion that this finding should be interpreted with caution since the initial hypothesis wanted the hippocampus to carry both traces. Next studies should use the lateral prefrontal cortex as a ROI and replicate this result for true recognition to add robustness to the current findings. Please elaborate this in the discussion.

Reviewer #2 (Remarks to the Author):

This is the second time I am reviewing this manuscript. The authors have done a good job of addressing my concerns, I have no further comments. I have confidence in the analysis, interpretation, and presentation of the data.

Reviewer #3 (Remarks to the Author):

The authors have addressed some of my initial concerns. As stated in my initial review, I believe the experimental design and the main results (Figure 3d) are quite interesting. While the revised manuscript is improved in some respect, I believe the presentation of results and interpretation is less compelling. Thus, I am left with a mixed impression. There are two points in particular that I raised before which have not been entirely addressed.

1. I still feel that results are over-interpreted in some places. I think it is important to use language that clearly distinguishes between what the data actually show versus the interpretation of those findings. Examples of overinterpretation include (but are not limited to):

- Line 393 "Our study found direct evidence for spontaneous reactivation ...". As I noted in my initial review and as Reviewer 1 noted, the evidence for reactivation is not direct. A reactivation account is a very reasonable interpretation, but it is an interpretation. The design of the study does not allow for a direct measure of reactivation. I do not personally feel that it is a fatal flaw that the study does not include a direct measure of reactivation. The key point is that the pattern similarity predicts memory outcomes, which I think most readily aligns with a reactivation interpretation.

- Line 399 "we found that when people read misinformation, the representation of corresponding original information is retrieved and revived ..." Again, the idea that the original representation is retrieved and revived is an interpretation.

- Line 414 "the representation of original information faded in the hippocampus ...". The idea of a "fading" representation goes beyond what the data show. I do not think the data particularly suggest this, but at a minimum

2. I still think the prefrontal results are not that intuitive and are over-interpreted. The idea that

source monitoring was engaged when false memory occurred feels opposite to what might be predicted (i.e., that source monitoring processes would prevent false memory). But there are also many other possible interpretations of the prefrontal activation. It just seems impossible to have a specific interpretation of the correlation with prefrontal cortex without any other converging evidence to narrow down the potential function.

Reply to Reviewer 1's comments:

Issue 1.1

Thank you for addressing my major concern and adding this as a limitation of the current study. I agree that the difference between true and false memory in the neural pattern similarity observed in the hippocampus (everything else being equal) is “likely” to be explained by a memory representation being reinstated rather than re-processing of scenes elicited by words. However, this conclusion cannot be taken for granted given that it is always possible that by chance we observe greater or smaller neural pattern similarity for different conditions. For example, how do we interpret the greater than zero neural pattern similarity in the correct control condition within the hippocampus? Is that due to re-processing of the original scene or is it a memory effect?

Response:

We thank Reviewer 1 for his/her helpful suggestions. We agree with this reviewer that our main result (i.e., item-specific neural pattern similarity between post-event and memory-test stages in the hippocampus was higher for false memory than for true memory) provides strong, but not definitive, evidence for the reinstatement of memory representations. It is not definitive because, as we acknowledge in the discussion section, there is some perceptual overlap between stages in this study. However, Reviewer 1's new point about the greater-than-zero neural pattern similarity in the hippocampus for the correct controls does not necessarily suggest that perceptual overlap is the only explanation. It is likely that because the control items also shared specific details (e.g., a little girl with a ponytail and an old lady walking together), these details (rather than, or in addition to, perceptual overlap) would lead to item-specific neural pattern similarity between original-event and memory-test stages (i.e., OM) in the hippocampus. One piece of evidence supporting our speculation is that, in the hippocampus, the item-specific neural pattern similarity between original-event and memory-test stages was higher for correct controls than for incorrect controls (Supplementary Table S6). If the hippocampal pattern similarity were purely a reflection of perceptual reprocessing across stages, there should be no such a difference in control items, because for all memory types, participants underwent perceptual reprocessing of scenes across stages and saw all three options for each question in the memory test.

Issue 1.2

Thank you for performing an additional analysis. I would like to point out that the existence of the original and post-event memory traces in two different brain regions (lateral prefrontal cortex and hippocampus, respectively) is only exploratory since the former was only found after performing a searchlight analysis that was carried out subsequently. Thus, the authors should highlight in the discussion that this finding should be interpreted with caution since the initial hypothesis wanted the hippocampus to carry both traces. Next studies should use the lateral prefrontal cortex as a ROI and replicate this result for true recognition to add robustness to the current findings. Please elaborate this in the discussion.

Response:

Following Reviewer 1's helpful suggestion, we added the following sentence to the discussion.

Discussion, page 30, lines 535~537:

"In addition, it should be noted that the results of the cortical representation were based on the exploratory searchlight analysis, which should be replicated in future studies using ROI analysis."

Reply to Reviewer 2's comments:

This is the second time I am reviewing this manuscript. The authors have done a good job of addressing my concerns, I have no further comments. I have confidence in the analysis, interpretation, and presentation of the data.

Response:

We thank Reviewer 2 for his/her very encouraging comments.

Reply to Reviewer 3's comments:

The authors have addressed some of my initial concerns. As stated in my initial review, I believe the experimental design and the main results (Figure 3d) are quite interesting. While the revised manuscript is improved in some respect, I believe the presentation of results and interpretation is less compelling. Thus, I am left with a mixed impression. There are two points in particular that I raised before which have not been entirely addressed.

1. I still feel that results are over-interpreted in some places. I think it is important to use language that clearly distinguishes between what the data actually show versus the interpretation of those findings. Examples of overinterpretation include (but are not limited to):
- Line 393 "Our study found direct evidence for spontaneous reactivation ...". As I noted in my initial review and as Reviewer 1 noted, the evidence for reactivation is not direct. A reactivation account is a very reasonable interpretation, but it is an interpretation. The design of the study does not allow for a direct measure of reactivation. I do not personally feel that it is a fatal flaw that the study does not include a direct measure of reactivation. The key point is that the pattern similarity predicts memory outcomes, which I think most readily aligns with a reactivation interpretation.

- Line 399 "we found that when people read misinformation, the representation of corresponding original information is retrieved and revived ...". Again, the idea that the original representation is retrieved and revived is an interpretation.

- Line 414 "the representation of original information faded in the hippocampus ...". The idea of a "fading" representation goes beyond what the data show. I do not think the data particularly suggest this, but at a minimum.

Response:

We thank Reviewer 3 for these suggestions. In this revision, we corrected the over-interpretations and made the following revisions accordingly. Examples are as follows.

- Line 393 (In the revision: Discussion pages 24~25, lines 412~415). We deleted the assertion about "direct evidence". The new sentence is as follows:

“Extending previous univariate studies, our study found hippocampal pattern similarity between original and post-event stages, which suggests spontaneous reactivation of corresponding original information in the hippocampus during the post-event misinformation stage.”

- Line 399 (In the revision: Discussion page 25, lines 418~420). We changed “found” to “suggests”, and “is retrieved....” to “could be retrieved...”. The new sentence is as follows:

“On the contrary, our finding suggests that when people read misinformation, the representation of the corresponding original information could be retrieved and revived in the hippocampus.”

- Line 414 (In the revision: Discussion pages 25~26, lines 433~436). We deleted “faded” and separated the statements into the finding and the interpretation. “When participants took the memory test one day later, the representation of original information was found to be weakened in the hippocampus for true memory, whereas the hippocampus seemed to reinstate both original information and post-event (mis)information for false memory and correct control.”

2. I still think the prefrontal results are not that intuitive and are over-interpreted. The idea that source monitoring was engaged when false memory occurred feels opposite to what might be predicted (i.e., that source monitoring processes would prevent false memory). But there are also many other possible interpretations of the prefrontal activation. It just seems impossible to have a specific interpretation of the correlation with prefrontal cortex without any other converging evidence to narrow down the potential function.

Response:

In this revision, we corrected the over-interpretation and added the following sentence in the discussion.

Discussion, pages 29~30, lines 514~520:

“However, it should be noted that the hippocampal-prefrontal correlation found in this study was based on the exploratory searchlight analysis and is subject to alternative interpretations (e.g., bidirectional prefrontal-hippocampal interactions). Future studies should replicate this finding and investigate the causal role of prefrontal activation by using behavioral interventions (e.g., warning) or brain stimulation (e.g., transcranial direct current stimulation over the prefrontal cortex).”